# HETerogeneous vectorized or Parallel (HETPv1.0): An updated inorganic heterogeneous chemistry solver for the metastable state $NH_4^+$–$Na^+$–$Ca^{2+}$–$K^+$–$Mg^{2+}$–$SO_4^{2-}$–$NO_3^-$–$Cl^-$–$H_2O$ system based on ISORROPIA II

Stefan J. Miller[1], Paul A. Makar[1], Colin J. Lee[1]

[1]Air Quality Modelling and Integration Section, Air Quality Research Division, Atmospheric Science and Technology Directorate, Environment and Climate Change Canada, 4905 Dufferin Street, Toronto, Ontario, M3H 5T4, Canada

*Correspondence to*: Stefan Miller (Stefan.Miller@ec.gc.ca), Paul Makar (Paul.Makar@ec.gc.ca)

**Abstract.**

We describe a new Fortran computer program to solve the system of equations for the $NH_4^+$–$Na^+$–$Ca^{2+}$–$K^+$–$Mg^{2+}$–$SO_4^{2-}$–$NO_3^-$–$Cl^-$–$H_2O$ system, based on the algorithms of ISORROPIA II. Specifically, the code solves the system of equations describing the "forward" (gas + aerosol input) metastable state, but with algorithm improvements and corrections. These algorithm changes allow the code to deliver more accurate solution results in formal evaluations of accuracy of the roots of the systems of equations, while reducing processing time in practical applications by about 50 %. The improved solution performance results from several implementation improvements relative to the original ISORROPIA algorithms. These improvements include (i) the use of the 'interpolate, truncate and project' (ITP) root–finding approach rather than bisection, (ii) the allowance of search interval endpoints as valid roots at the onset of a search, (iii) the use of a more accurate method to solve polynomial subsystems of equations, (iv) the elimination of negative concentrations during iterative solutions, (v) corrections for mass conservation enforcement, and (vi) several code structure improvements. The new code may be run in either a "vectorization" mode wherein a global convergence criterion is used across multiple tests within the same chemical subspace, or a "by case-by-case" mode wherein individual test cases are solved with the same convergence criteria. The latter approach was found to be more efficient on the compiler tested here, but users of the code are recommended to test both options on their own systems. The new code has been constructed to explicitly conserve the input mass for all species considered in the solver, and is provided as open source Fortran shareware.

## 1 Introduction

Anthropogenic atmospheric particulate matter (aerosols) can negatively impact the Earth's climate and biosphere – aerosols can alter the atmosphere's radiative forcing (Jacobson, 2001; Schmale et al., 2021), contribute to acid rain (Irwin and Williams, 1988), reduce atmospheric visibility (Quan et al., 2015) and cause morbidity in humans (Atkinson et al., 2014) and other plant and animal species (Lovett et al., 2009). Atmospheric particulate matter is comprised of organic and inorganic species, with

25 to 60 % of particulate matter being inorganic by mass (Harrison and Pio, 1983; Heintzenberg, 1989). The inorganic portion of atmospheric particulate matter consists primarily of sulfate ($SO_4^{2-}$), nitrate ($NO_3^-$), ammonium ($NH_4^+$), chloride ($Cl^-$), calcium ($Ca^{2+}$), potassium ($K^+$), magnesium ($Mg^{2+}$), sodium ($Na^+$) and water ($H_2O$) (Harrison and Pio, 1983; Wang et al., 2003). Along coastlines and within marine air masses, inorganic bromide ($Br^-$) (Sander et al., 2003) and iodide ($I^-$) (Saiz-Lopez et al., 2011) may also be common. $Ca^{2+}$, $K^+$, $Mg^{2+}$, $Na^+$ and $Cl^-$ exist principally in coarse mode aerosols (particle

diameter $> 2.5$ µm), and these species are particularly important to the partitioning of ammonium and nitrate (Metzger et al., 2006). As an example, coarse mode particle nitrate may form via adsorption of nitric acid ($HNO_3$) onto sea salt (Savoie and Prospero, 1982). It should be noted that a considerable amount of $K^+$ may also be present in fine mode aerosols (particle diameter $< 2.5$ µm) when it is generated during biomass burning events, termed 'pyrogenic potassium' (Metzger et al., 2006). The transfer of cation and anion mass between the gas and particulate phase is crucially dependent on inorganic thermodynamic

partitioning. For example, observations have indicated that base cations ($Ca^{2+}$, $K^+$, $Mg^{2+}$, $Na^+$) and $NH_4^+$ can compete for uptake of $HNO_3$ (the former residing in coarse mode, the latter in fine mode particle nitrate formation) (Makar et al., 1998, Anlauf et al., 2006).

The aerosols can reside in the crystalline solid phase or exist as an aqueous solution of ions and may be in thermodynamic equilibrium with atmospheric gases. The partitioning of the inorganic species between the solid, gaseous, and

aqueous phase is a complex computational problem, owing to the many nonlinearities involved. The equations describing high concentration (non-ideal) inorganic heterogeneous equilibrium between gases, ions and crystallized solid phases present a system of $N$ equations in $N$ unknowns, where $N$ is the number of chemical constituents. While these equations may be addressed through searching for roots of polynomials resulting from substitution of equations, the non-ideal nature of the problem manifests as corrections to the equilibrium constants in the equations, known as activity coefficients. The activity

coefficients depend on concentrations in the condensed phase, increasing the nonlinearity of the system of equations and requiring the development of special techniques for their solution. Several solvers have been developed to simulate the thermodynamic partitioning of inorganic species (see Zhang et al., 2000 and Pye et al., 2020 for a detailed review of these solvers). AIM2 (Clegg and Pitzer, 1992; Wexler and Clegg, 2002), GFEMN (Ansari and Pandis, 1999a, b) and UHAERO (Amundson et al., 2006) are considered the most rigorous solvers, in that they attempt to find a global minimum in the Gibbs

free energy of the constituents. However, the downfall of this approach stems from the computational time and operator review required to discriminate between the true global minimum and potentially many local minima (Makar et al., 2003). This difficulty has prevented the use of these solvers in three dimensional (3D) chemical transport models (CTMs) to date. However, these models may be used to help determine subsystems of equations – local solution spaces where gas and aerosol partitioning will occur with a smaller number of constituents – and hence describe simplified systems that may be solved with

more efficient methods. Inorganic heterogeneous chemistry implementations in CTMs have relied on computationally efficient algorithms. These algorithms directly solve the system of inorganic heterogeneous chemistry equations by considering the species' chemical potentials within these predetermined subspaces of a smaller numbers of species, hence simplifying and reducing the number of equations and unknowns. The specific subspace to be solved is determined based on

the input precursor species, and the ratio(s) of the total available cations to the total available sulfate (see Sect. 2). This approach effectively breaks the larger problem into several separate smaller problems. Solvers that apply this tactic include SCAPE (Kim et al., 1993a,b; Kim and Seinfeld, 1995; Meng et al., 1995), EQUILSOLV–II (Jacobson, 1999), ISORROPIA/ISORROPIA II/ISORROPIA–lite (Nenes et al., 1998; Fountoukis and Nenes, 2007; Kakavas et al., 2022), HETV (Makar et al., 2003) and HETP (presented herein). HETV (HETerogeneous Vectorized) was a vectorized solver (i.e., optimized for vectorized computer architecture) based on the original ISORROPIA algorithms (Nenes et al., 1998), but with numerical improvements related to more accurate evaluation of cubic and quadratic equations whose coefficients may vary by several orders of magnitude, coding structure changes to replace logical IF statements with mathematical equivalents, the elimination of redundant calculations, the replacement of intrinsic functions in activity coefficient calculations by high order Taylor series, and the gathering of similar problems within a single-subsystem to be solved using a global convergence criteria. These modifications allowed HETV to perform calculations in 1/38 to 1/89 of the time required for ISORROPIA (v1.0), on a vector supercomputer (the fastest supercomputer architecture at the time the HETV code was created). More recent supercomputer architectures focus on parallel processing across multiple processors to reduce processing time. In 2007, an update to ISORROPIA was released that included 'crustal' species ($Mg^{2+}$, $K^+$, $Ca^{2+}$) and sea salt ($Na^+$, $Cl^-$), referred to as ISORROPIA II (Fountoukis and Nenes, 2007). More recently, a simplified and extended version of ISORROPIA II has been developed, called ISORROPIA-lite. ISORROPIA-lite addresses the metastable state (i.e., it assumes a supersaturated aqueous solution where crystalline states are ignored, except $CaSO_4$), as well as effects of organic aerosols on the partitioning of the inorganic system. ISORROPIA-lite solves the same chemical subspaces as ISORROPIA II, but only for the metastable state option and uses precalculated binary activity coefficients, resulting in a solver that executes about 35 % faster than ISORROPIA II (Kakavas et al., 2022).

The underlying issue driving the use of a metastable state assumption in regional air quality models for inorganic heterogeneous chemistry solvers is that the presence of water in the aerosol is not only controlled by the inorganic components, but also by other components within a mixed-phase aerosol. In the absence of these additional sources of aerosol water, the "pure" (i.e. only) inorganic aerosol thermodynamics can result in partitioning to the 'stable' aerosol phase as only crystalline salts (no ions) or a mixture of crystalline salts and aqueous ions that are saturated with respect to the crystalline salts. The presence of the additional sources of aerosol water will ensure that some water is always present – and hence the subsystems of equations that have no water will not be encountered. It has been reported that metastable state aerosols may be 'ubiquitous' in the Earth's atmosphere, existing more than 50 % of the time when the relative humidity is between 45 and 75 % (Rood et al., 1989; Tang et al., 1995); this may be especially true in the case of dissolved impurities such as organic species (Fountoukis et al., 2009). Another issue driving the use of the metastable state assumption in regional air quality models is the need to track the RH history of aerosols to accurately predict their phase state, due to the hysteresis of salts. Specifically, without knowing the RH history of the aerosol, it is not possible to determine whether the aerosol will exist as an aqueous solution of ions or as a crystalline salt between its efflorescence and deliquescence RH (Martin et al., 2004; Fountoukis et al., 2009). Given these reasons, applications of inorganic aerosol thermodynamics within CTMs tend to assume a metastable state as the

most likely conditions in the troposphere, although absolutely stable aqueous aerosols are possible above the deliquescence RH. This assumption also reduces the number of chemical subspaces required to obtain a solution of the system of equations

100 for inorganic heterogeneous chemistry, and additions such as formulae for the water activity associated with organic aerosols may be used to better simulate the aerosol water content (Kakavas et al, 2022).

   In the different versions of ISORROPIA and HETV, the roots of subsystems of equilibrium equations are used to determine the thermodynamic equilibrium solution, the result being the concentrations of the inorganic ions and the partitioning gases. In ISORROPIA/ISORROPIA II/ISORROPIA–lite and HETV, convergence of these solutions to these

105 systems of equations are obtained via a bisection search, while in SCAPE, Newton's method is employed. Newton's method is also used in ANISORROPIA where it is combined with the bisection method for chemical subspaces describing a neutral aerosol. ANISORROPIA performs a sensitivity analysis on each inorganic species considered in ISORROPIA (excluding $Ca^{2+}$, $Mg^{2+}$, $K^+$) with respect to the total input precursor species concentration (Capps et al., 2012). It is well known that Newton's method may fail to converge if the 'initial guess' of the root is too far away from the actual root (Burden and Faires,

110 2011). Unlike Newton's method, the bisection method is guaranteed to converge, although the convergence may be slow. The bisection method requires at most $n_{eval} = \log_2\left(\frac{b-a}{2\varepsilon}\right)$ function evaluations to locate the root ($x$) on the interval $[a, b]$ such that $|x_i - x^*| \leq \varepsilon$, where $\varepsilon$ is a set tolerance, $x^*$ is the current estimate of the root, and $x_i$ is the previous estimate of the root. In most cases, the bisection method will require all $n_{eval}$ function evaluations for convergence (Oliveira and Takahasi, 2021). Recently, Oliveira and Takahasi (2021) developed a *modified* bisection approach called "interpolate, truncate and project"

115 (ITP), which may obtain superlinear convergence, therefore reducing the execution time required to obtain a solution with the same accuracy as the typical bisection method (note that the bisection method has linear convergence). To achieve an improved order of convergence, the ITP method incorporates a regula-falsi estimate into the bisection method. The 'typical' bisection method simply splits the original interval in half, with $x^*$ becoming the midpoint of this interval ($x^* = x_{1/2} = 0.5(a + b)$); a new interval is then chosen (i.e., $[a, x^*]$ or $[x^*, b]$) based on the sign change. The regula-falsi estimate, however, is determined

120 by fitting a straight line through the identified interval by using the function values at each endpoint (i.e., $x_f = [bf(a) - af(b)]/[f(a) - f(b)]$). This estimate defines the 'interpolation' aspect of the ITP method. By making use of these two estimates simultaneously (i.e., $x^*$ and $x_f$), ITP can outperform the typical bisection method for both convergence rate and accuracy. For well-behaved functions (i.e., there is exactly one root in the function's domain) ITP requires on average 24 to 37 % of the iterations required by bisection, and for ill-behaved functions (i.e., there are multiple roots in the function's domain

125 or the function contains discontinuities) ITP requires on average 82 % of the iterations required by bisection. Oliveira and Takahasi (2021) also compared the performance of ITP against well-established alternative root-finding methods, such as Ridder's method, the Illinois method, Matlab's 'fzero' routine and the Secant method. For all mathematical functions evaluated for convergence, ITP required the least amount of function evaluations when compared against the other root-finding methods. For example, compared to Ridder's method, ITP requires an average of 20.2 function evaluations while Ridder's

method requires an average of 26.1. The full mathematical details describing the ITP method (as well as pseudocode) are given in Oliveira and Takahasi (2021) and are not repeated herein.

    In this work we present HETP (HETerogeneous vectorized or Parallel), a solver based on the "forward" (input precursor species as gas + aerosol) metastable state algorithms of ISORROPIA II. HETP has been optimized for vector processors where similar problems for a subsystem are gathered and solved with a global convergence criterion, or parallel

processors, where local case-by-case solutions to the system of equations are used to minimize processing time. HETP focuses exclusively on the metastable state where some amount of liquid water is always assumed to be present in the aerosol, even at very low relative humidity. The metastable state assumption is currently applied in various state-of-the-art global and regional CTMs, such as GEM-MACH, GEOS-Chem and CMAQ. GEM-MACH uses HETV (Makar et al., 2018), while CMAQ (Wang et al., 2012) and GEOS-Chem (Pye et al., 2009) use ISORROPIA II. HETP has been updated to improve its numerical stability

and computational speed compared to ISORROPIA II, as will be discussed in detail below. Specifically, in addition to the numerical improvements associated with its predecessor, HETV, modifications have been made to incorporate base cations and chlorine, to ensure mass conservation, and to update the bisection method to ITP. In the following sections, we demonstrate that the implementation of ITP not only decreases the execution time of the solver, but it can also improve the final convergence of the chemical system by initializing the search with a species concentration (i.e., an initial guess) that is

closer to the actual solution being sought at thermodynamic equilibrium. Thus, we have developed a new solver (HETP) that has improved the accuracy and decreased the execution time compared to the original ISORROPIA II metastable state forward algorithms. Section 2 briefly outlines the background theory underpinning the solver, followed in Sect. 3 by a detailed list of modifications that are unique to HETP (relative to ISORROPIA II). The final sections provide a comprehensive comparison between ISORROPIA II and HETP, in terms of output results and computational speed, both of which are improved in the

HETP algorithm. For brevity we will henceforth refer to ISORROPIA II as ISORROPIA in the remainder of this paper.

## 2 Background theory

HETP is based on the algorithms of ISORROPIA, which are in turn based on Gibbs free energy minimizations to define subspaces of systems of equations for inorganic heterogeneous chemistry. ISORROPIA solves two types of problems, referred to as the 'forward' or 'reverse' problem. The forward problem requires known input precursor concentrations (total gas +

aerosol), along with a relative humidity and air temperature, to predict the equilibrium state. HETP does not consider the reverse problem where the relative humidity, air temperature and aqueous aerosol species concentrations are known (i.e., no gaseous species are included in the input precursor concentrations), and a solution is sought to determine the resulting equilibrium and gas concentrations. For measured data, the reverse problem is typically not recommended since it lacks the inclusion of gas phase speciation in the input, making its predictions highly sensitive to measurement errors. For example,

Hennigan et al., (2015) show that a $\pm 10$ % measurement error in $NH_4^+$ can alter the pH predicted by the reverse mode by more than 1 pH unit. Furthermore, Song et al., (2018) found that the aerosol pH predicted by the reverse mode may result in a

bimodal pH distribution; in their study a negative ion balance gave highly acidic conditions while a positive ion balance gave near-neutral conditions. We note that the reverse mode is used in CMAQ to perform mass transfer with the coarse mode (Pye et al., 2022), but other CTMs that employ ISORROPIA use only the forward mode.

The ISORROPIA solvers have been used in a large number of CTM applications (i.e., ISORROPIA: 1250 citations; ISORROPIA II: 1245 citations), and have been a key component in these models, allowing inorganic heterogeneous chemistry calculations to be carried out in a timely fashion. Here, we build on those solvers, and would like to acknowledge their important contribution to air-quality modelling science. As stated in Sect. 1, HETP assumes a metastable state, where some liquid water is always present even at low relative humidity. The required input precursor species are total sulfate (TS,

expressed as molar equivalent $H_2SO_4$), total ammonium (TA, expressed as molar equivalent $NH_3$), total nitrate (TN, expressed as molar equivalent $HNO_3$), total sodium (TNa, expressed as molar equivalent $Na^+$), total chloride (TCl, expressed as molar equivalent HCl), total magnesium (TMg, expressed as molar equivalent $Mg^{2+}$), total potassium (TK, expressed as molar equivalent $K^+$) and total calcium (TCa, expressed as molar equivalent $Ca^{2+}$). Units of these net precursor species are mol m$^{-3}$ air upon input into both ISORROPIA and HETP. For some input conditions ISORROPIA will adjust the input precursor

concentrations prior to determining the subroutine that should be entered. Specifically, ISORROPIA will adjust TA and TCl so that they are no less than $1\times10^{-10}$ mol m$^{-3}$, and if (TNa + TS + TN) < $1\times10^{-10}$ mol m$^{-3}$, then ISORROPIA will adjust TNa and TN so that they are no less than $1\times10^{-10}$ mol m$^{-3}$ (note these are applicable only to Branch 3 and 4; see Fig. 1). These adjustments performed within a CTM result in output speciation that violates mass conservation, since mass is created for TA, TN, TCl and TNa. For example, for 50,000 unique sets of input conditions executing Branch 4 subroutines (i.e., winter input

from Sect. 4.2), performing these adjustments results in a median of $1.09\times10^{-3}$ ug m$^{-3}$ of TCl being created by the solver. On a relative scale $\left(\frac{\text{output mass}}{\text{input mass}}\times 100\ \%\right)$ this represents a median increase in TCl mass by 42.7 %; for 25 % of these input conditions the relative increase in TCl mass $\geq$ 4414 %. In a CTM these mass violations would occur at a single timestep, therefore the impact would increase as the simulation progresses. Considering these mass violations, ISORROPIA currently used in GEOS-Chem v14.0.0 (GEOS-Chem, 2022) does not perform these mass adjustments. It should be noted that GEOS-

Chem v14.0.0 uses ISORROPIA v2.2 which contains minor bug fixes compared to ISORROPIA II (v2.0). CMAQv5.4 which also uses ISORROPIA v.2.2 (CMAS, 2016; USEPA, 2022) does perform these initial mass adjustments; however, any output that results from input data that are mass adjusted are flagged. HETP adopts the approach of GEOS-Chem and likewise does not perform these initial mass adjustments. Therefore, ISORROPIA v2.2 used herein (obtained from CMAQv5.4; USEPA, 2022) has been modified so that it also does not perform the aforementioned mass adjustments. Other than this modification,

the branches and chemical subspaces shown in Fig. 1 are identical to ISORROPIA.

Table 1 lists the entire set of equilibrium reactions (ER1 to ER7) that are solved in various chemical subspaces of the metastable state 'forward' option of both ISORROPIA and HETP. The decision tree (outlined at the end of this section) used to select the appropriate chemical subspace, as well as the equilibrium reactions shown in Table 1, are identical to ISORROPIA (Fountoukis and Nenes, 2007). ER1 to ER7 are solved by introducing additional relationships for mass conservation,

electroneutrality (i.e., a charge balance equation), aerosol water activity, and mean molality-based activity coefficients ($\gamma$) to represent ion-ion interactions in non-ideal aqueous solutions ($\gamma \to 1$ as the solution becomes more dilute, i.e. more "ideal"). Given in Table S1 are the equilibrium reactions that form the basis of dry salt partitioning (ER8 to ER25) that is completed during the initialization of several metastable state subspaces. It should be noted that ER8 to ER25 are not solved directly – instead the input precursor species are partitioned into various salts based on these equilibrium reactions.

The exact salts that form (i.e., which anions are matched by which cations) depends on the specific chemical subspace that is entered and whether the subspace is 'sulfate rich', 'sulfate super-rich' or 'sulfate poor'; these classifications are determined by the relative amounts of the input cations to the total available sulfate. For example, in CALCP13 (the algorithm branch describing a sulfate poor case with base cations present) calcium, potassium and magnesium first react with the sulfates to produce $CaSO_4$, $K_2SO_4$ and $MgSO_4$ respectively, and sodium and chloride react to form NaCl. Any free calcium will then

react with nitrate and free chloride to form $Ca(NO_3)_2$ and $CaCl_2$ respectively. Next, free magnesium will react with free nitrate and free chloride to form $Mg(NO_3)$ and $CaCl_2$, respectively, and then free sodium will react with free nitrate to form $NaNO_3$. Finally, free potassium will react with free chloride and free nitrate to form KCl and $KNO_3$, respectively. The order of dry salt partitioning in the remaining chemical subspaces, where applicable, are provided in Table S2 of the Supplemental Information and are identical to ISORROPIA (except for CALCL9, discussed in Sect. 3). Depending on the amount of anions and cations

present for this initial partitioning stage, some of these input components may be in excess of the amount which can be partitioned into salts. This excess mass, beyond that required to create a set of salts, is referred to as the "free" amount of the given component. The salts created in this initial stage of partitioning are then assumed to undergo deliquescence in each of the problems to be solved, resulting in aqueous phase speciation that is then used as the initial conditions for which a thermodynamic solution is required. In addition to the free amounts generated within a chemical subspace during dry salt

partitioning, free amounts may also be generated during the initialization of HETP and ISORROPIA, prior to entering a chemical subspace. Specifically, automatic adjustments are applied if the input precursor species are nonelectroneutral. In this case, any excess cations are ignored, and free amounts of Na, Ca, K and Mg may be created. These automatic adjustments help constrain the particle alkalinity of the equilibrium solution, ensuring that it does not exceed the pH of dissolved particulate calcium carbonate (Pye et al., 2020). The "free" mass must therefore be treated carefully in the context of the *application* of

thermodynamic solvers within CTMs. A key requirement for CTMs is that they conserve the mass of transported species, within process representation such as inorganic thermodynamics. Solvers such as ISORROPIA conserve mass for the "captured" or "non-free" portion of the input chemical speciation, but not the "free" portion. Currently, ISORROPIA only outputs the aqueous, solid, or gaseous species that result after partitioning at thermodynamic equilibrium and not 'free' amounts. If the non-volatile species ($Ca^{2+}$, $Mg^{2+}$, $K^+$, $Na^+$) output by the solver are used by the CTM, and the 'free' amounts

are not retained and used to conserve mass, then inputs to the solver which result in 'free' species will be lost in the solver call. We note, however, that CTMs such as CMAQ v5.4 and GEOS–Chem v14.0.2 avoid this potential problem by only allowing the semi-volatile species (i.e., $Cl^-$–HCl, $NO_3^-$–$HNO_3$, $NH_4^+$–$NH_3$) to be modified on output from the solver. The semi-volatile species are then saved and transferred back to the model. The non-volatile species are not used after chemical partitioning and

are not transferred back to the model calling ISORROPIA. Therefore, any non-volatile free mass that was created in
ISORORPIA is not lost in the solver call in these CTMs (aerosol mass is conserved). In HETP the free amounts have been
retained in all cases and are returned to the calling code for completeness. The manner in which the initial salt concentrations
are determined, including the "free" amounts, is provided in detail in Table S2 (Supplemental information). HETP tracks all
free amounts explicitly, otherwise, the initial dry salt concentrations outlined in Table S2 are determined identical to
ISORROPIA (except CALCL9 which is discussed in Sect. 3).


**Table 1:** Equilibrium reactions (ER) considered in the metastable state chemical subspaces (Fountoukis and Nenes, 2007). These reactions are solved directly within the appropriate major system. $\Delta G_f^0$, $\Delta H_f^0$ and $\Delta C_p^0$ are the standard molar Gibbs free energy, enthalpy of formation and heat capacity at standard pressure, $R = 8.314$ J mol$^{-1}$ K$^{-1}$ is the universal gas constant, and $T_0 = 298.15$ K is the reference temperature. The ions denoted in square brackets [...] (i.e., [H$^+$], [SO$_4^{2-}$], [HSO$_4^-$], etc.) refer to molalities with units of mol kg$^{-1}$. Here, $\gamma$ is a multicomponent activity coefficient and $p$ is a gas partial pressure.
Theoretically, equilibrium constants are unitless since each pressure or concentration should be normalized by a standard state; here standard states are neglected.

| Equation No. | Equilibrium reactions and values of $\exp\left(-\Delta G_f^0/(RT_0)\right), -\Delta H_f^0/(RT_0), -\Delta C_p^0/R$ | | Equilibrium equation |
|---|---|---|---|
| **ER1** | $K_{\mathrm{HSO_4}}: \mathrm{HSO_4^-}_{(aq)} \rightleftharpoons \mathrm{H^+}_{(aq)} + \mathrm{SO_4^{2-}}_{(aq)}$ | | $K_{\mathrm{HSO_4}} = \frac{[\mathrm{H^+}][\mathrm{SO_4^{2-}}]}{[\mathrm{HSO_4^-}]}\left(\frac{\gamma_{\mathrm{H^+}}\gamma_{\mathrm{SO_4^{2-}}}}{\gamma_{\mathrm{HSO_4^-}}}\right)$ [mol kg$^{-1}$] |
| | $\exp\left(-\Delta G_f^0/(RT_0)\right)$ | $1.015\times10^{-2}$ | |
| | $\Delta H_f^0/(RT_0)$ | 8.85 | |
| | $\Delta C_p^0/R$ | 25.14 | |
| **ER2** | $K_{\mathrm{NH_3}_a}: \mathrm{NH_3}_{(g)} \rightleftharpoons \mathrm{NH_3}_{(aq)}$ | | $K_{\mathrm{NH_3}_a} = \frac{[\mathrm{NH_3}_{(aq)}]}{[p_{\mathrm{NH_3}_{(aq)}}]}\left(\gamma_{\mathrm{NH_3}_{(aq)}}\right)$ [mol kg$^{-1}$ atm$^{-1}$] |
| | $\exp\left(-\Delta G_f^0/(RT_0)\right)$ | $5.7639\times10^{1}$ | |
| | $\Delta H_f^0/(RT_0)$ | 13.79 | |
| | $\Delta C_p^0/R$ | −5.39 | |
| **ER3** | $K_{\mathrm{NH_3}_b}: \mathrm{NH_3}_{(aq)} + \mathrm{H_2O}_{(aq)} \rightleftharpoons \mathrm{NH_4^+}_{(aq)} + \mathrm{OH^-}_{(aq)}$ | | $K_{\mathrm{NH_3}_b} = \frac{[\mathrm{NH_4^+}][\mathrm{OH^-}]}{[\mathrm{NH_3}_{(aq)}]a_w}\left(\frac{\gamma_{\mathrm{NH_4^+}}\gamma_{\mathrm{OH^-}}}{\gamma_{\mathrm{NH_3}_{(aq)}}}\right)$ [mol kg$^{-1}$] |
| | $\exp\left(-\Delta G_f^0/(RT_0)\right)$ | $1.805\times10^{-5}$ | |
| | $\Delta H_f^0/(RT_0)$ | −1.50 | |
| | $\Delta C_p^0/R$ | 26.92 | |
| **ER4** | $K_{\mathrm{H_2O}}: \mathrm{H_2O}_{(aq)} \rightleftharpoons \mathrm{H^+}_{(aq)} + \mathrm{OH^-}_{(aq)}$ | | $K_{\mathrm{H_2O}} = \frac{[\mathrm{H^+}][\mathrm{OH^-}]}{a_w}\left(\gamma_{\mathrm{H^+}}\gamma_{\mathrm{OH^-}}\right)$ [mol$^2$ kg$^{-2}$] with $\gamma_{\mathrm{H^+}} = 1$ and $\gamma_{\mathrm{OH^-}} = 1$ |
| | $\exp\left(-\Delta G_f^0/(RT_0)\right)$ | $1.010\times10^{-14}$ | |
| | $\Delta H_f^0/(RT_0)$ | −22.52 | |
| | $\Delta C_p^0/R$ | 26.92 | |
| **ER5** | $K_{\mathrm{HNO_3}}: \mathrm{HNO_3}_{(g)} \rightleftharpoons \mathrm{H^+}_{(aq)} + \mathrm{NO_3^-}_{(aq)}$ | | $K_{\mathrm{HNO_3}} = \frac{[\mathrm{H^+}][\mathrm{NO_3^-}]}{p_{\mathrm{HNO_3}}}\left(\gamma_{\mathrm{H^+}}\gamma_{\mathrm{NO_3^-}}\right)$ [mol$^2$ kg$^{-2}$ atm$^{-1}$] |
| | $\exp\left(-\Delta G_f^0/(RT_0)\right)$ | $2.511\times10^{6}$ | |
| | $\Delta H_f^0/(RT_0)$ | 29.17 | |
| | $\Delta C_p^0/R$ | 16.83 | |
| **ER6** | $K_{\mathrm{HCl}}: \mathrm{HCl}_{(g)} \rightleftharpoons \mathrm{H^+}_{(aq)} + \mathrm{Cl^-}_{(aq)}$ | | $K_{\mathrm{HCl}} = \frac{[\mathrm{H^+}][\mathrm{Cl^-}]}{p_{\mathrm{HCl}}}\left(\gamma_{\mathrm{H^+}}\gamma_{\mathrm{Cl^-}}\right)$ [mol$^2$ kg$^{-2}$ atm$^{-1}$] |
| | $\exp\left(-\Delta G_f^0/(RT_0)\right)$ | $1.971\times10^{6}$ | |
| | $\Delta H_f^0/(RT_0)$ | 30.20 | |
| | $\Delta C_p^0/R$ | 19.91 | |
| **ER7** | $K_{\mathrm{NH_4NO_3}}: \mathrm{NH_4NO_3}_{(s)} \rightleftharpoons \mathrm{NH_3}_{(g)} + \mathrm{HNO_3}_{(g)}$ | | $K_{\mathrm{NH_4NO_3}} = p_{\mathrm{NH_3}}p_{\mathrm{HNO_3}}$ [atm$^2$] |
| | $\exp\left(-\Delta G_f^0/(RT_0)\right)$ | $4.199\times10^{-17}$ | |
| | $\Delta H_f^0/(RT_0)$ | −74.735 | |
| | $\Delta C_p^0/R$ | 6.025 | |

Note: $\frac{\gamma_{\mathrm{H^+}}\gamma_{\mathrm{SO_4^{2-}}}}{\gamma_{\mathrm{HSO_4^-}}} = \frac{\gamma_{\mathrm{H^+}}^2\gamma_{\mathrm{SO_4^{2-}}}}{\gamma_{\mathrm{H^+}}\gamma_{\mathrm{HSO_4^-}}} = \frac{\gamma_{\mathrm{H_2SO_4}}^3}{\gamma_{\mathrm{H-HSO_4}}^2}$ (Kim and Seinfeld, 1993b)

The equilibrium constants are calculated from the Van't Hoff equation, where $\Delta H^0(T_0)$ is approximated for a small
temperature range (Denbigh, 1981) as

$$K_j(T) = K_0 \exp\left[-\frac{\Delta H^0(T_0)}{RT_0}\left(\frac{T_0}{T} - 1\right) - \frac{\Delta c_p^0}{R}\left(1 + \ln\left(\frac{T_0}{T}\right) - \frac{T_0}{T}\right)\right], \tag{1}$$

where $K_0$ is the equilibrium constant at a reference temperature of $T_0 = 298.15$ K, $R = 8.314$ J mol$^{-1}$ K$^{-1}$ is the universal gas constant, $\Delta c_p^0$ (J mol$^{-1}$ K$^{-1}$) is the change of molar heat capacity of products minus reactants and $\Delta H^0$ (J mol$^{-1}$) is the enthalpy change of the reaction at temperature $T_0$ (K). $K_0$ is determined as

$$K_0(T_0) = \exp\left(-\frac{\Delta G_f^0}{RT_0}\right), \tag{2}$$

where $\Delta G_f^0$ (J mol$^{-1}$) is the standard molar Gibbs free energy of formation at $T_0$.

The mean activity coefficients are calculated following the same methodology as in ISORROPIA: multicomponent activity coefficients are calculated according to Bromley's formula (Bromley, 1973), binary activity coefficients are determined from the Kusik-Meissner relationship (Kusik and Meissner, 1978), and the temperature dependence of the multicomponent activity coefficients are calculated following Meissner and Peppas (1973). HETP (as in ISORROPIA) assumes that OH$^-_{(aq)}$ is small compared to other species, and hence it is not used in the calculation of ionic strength. HETP only allows on-line calculation of activity coefficients and does not use precalculated look-up tables.

Aerosol liquid water content in kg m$^{-3}$ air is calculated according to the Zdanovskii-Stokes-Robinson (ZSR) relation (Robinson and Stokes, 1965), as

$$W = \sum_i \frac{M_i}{m_i(a_w)}, \tag{3}$$

where $M_i$ is the concentration of species $i$ in mol m$^{-3}$ air and $m_i$ is the molality (mol kg$^{-1}$) of an aqueous solution of $i$ at the same water activity ($a_w$) as the mixture. It is assumed that there are negligible effects from droplet curvature (i.e., Kelvin effect), and that the growth of an aerosol by uptake of H$_2$O does not affect the ambient water vapor pressure (i.e., no effect on the ambient RH). Therefore, equilibrium between the vapor (gas) and liquid (aerosol) phase is assumed with $a_w = $ RH (Seinfeld and Pandis, 2016).

There are other simplifications and assumptions applied to the metastable state in HETP and ISORROPIA including: (i) sulfuric acid, sodium, magnesium, calcium and potassium are assumed to only exist in the aerosol phase (i.e., no sulfuric acid gas), (ii) calcium sulfate (CaSO$_4$) never dissolves and will only be present as a solid species, (iii) in cases that are sulfate rich (B4, C2, E4, F2, I6, J3, L9, K4), the ions NH$_4^+$, NO$_3^-$ and Cl$^-$ are "assumed to be minor species that do not significantly perturb the [thermodynamic] equilibrium" (Fountoukis and Nenes, 2007) – the partitioning problem to be solved for these ions in sulfate-rich cases is referred to as a "minor system". All minor systems are solved after convergence of the major system has been achieved. Practically, for point (iii) above, this implies that NO$_3^-$ and Cl$^-$ within the minor system will not affect the charge balance or the activity coefficients of the major system. The concentration of H$^+$ determined from the major system is

used as the basis to perform the partitioning between the aerosol and gas phase in the minor system(s), using the equilibrium reaction(s) in Table 1 which describe the minor system(s) to be solved.

        The system of equations and order of the operations to create a solution is identical between ISORROPIA and HETP using the same chemical subspaces. The subspace that will be entered, and therefore the speciation that will be present, is determined based on the input precursor species. If crustal species (TK, TMg and TCa), TNa and TCl are all near zero, then

the set of chemical subspaces reduces to those used in HETV (Makar et al., 2002) and the original release of ISORROPIA (Nenes et al., 1998). Both codes follow the same procedure, first creating three sulfate ratios: the "total sulfate ratio" ($R_1$), "crustal species and sodium ratio" ($R_2$) and "crustal species ratio" ($R_3$),

$$R_1 = \frac{\text{TA+TNa+TCa+TK+TMg}}{\text{TS}} \tag{4}$$


$$R_2 = \frac{\text{TNa+TCa+TK+TMg}}{\text{TS}} \tag{5}$$

$$R_3 = \frac{\text{TCa+TK+TMg}}{\text{TS}}. \tag{6}$$

These ratios are used as the basis to determine the appropriate chemical subspace that is entered and solved, with 15 possible metastable subspaces in total. The possible subspaces given the input ratios $R1$, $R2$ and $R3$ are summarized in Fig. 1, along with the resulting speciation (aqueous, gaseous, and solid). The bold font species are solved in the major system while regular font species are solved in the minor system. Four unique 'branches' exist: in Branch 1 only TS and TA are present, in Branch 2 only TS, TA and TN are present, in Branch 3 TS, TA and TN are present, and at least one of TNa or TCl, and in Branch 4

TS, TA and TN are present, and at least one of TCa, TK or TMg. The branches are further subdivided into subcases depending on input concentrations. It should be noted that the subcases G5, H6, O7, M8 and P13 require that TCl be present, along with the aforementioned requisite species, otherwise a solution is not possible due to small numbers and floating-point arithmetic limitations. This limitation occurs since HETP does not apply the mass modification that resets TCl to a floor value of $1 \times 10^{-10}$ mol m$^{-3}$, as discussed near the start of the section.


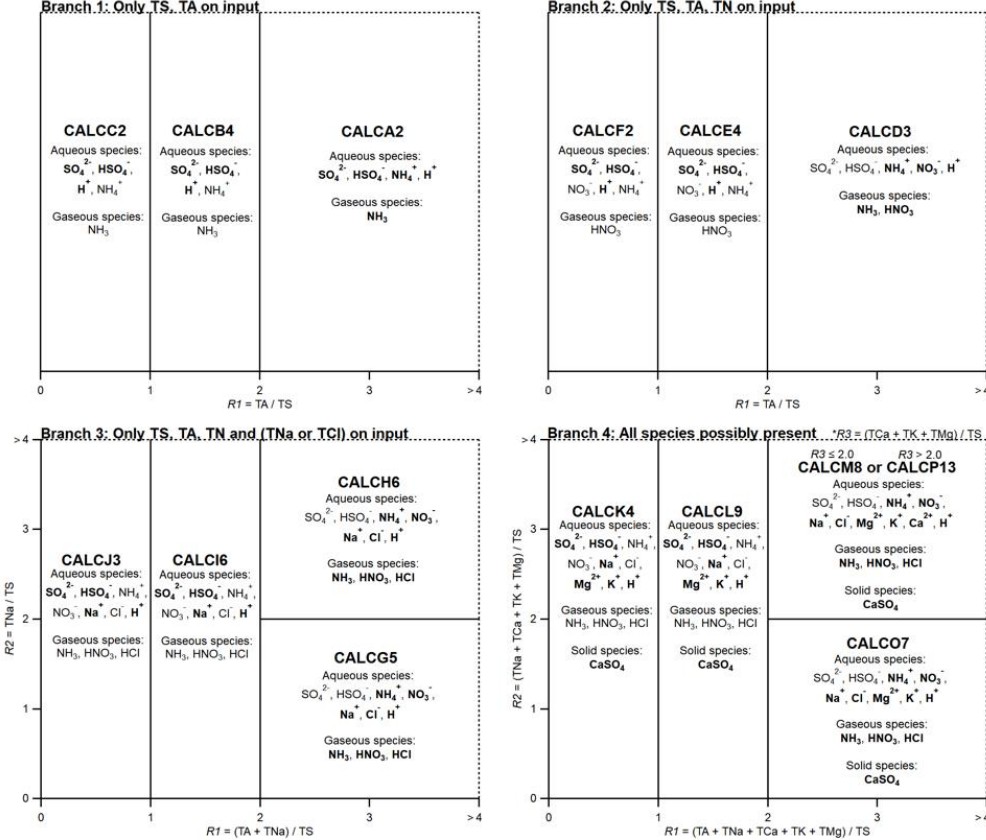

**Figure 1**: Domains of the systems of equations, based on ISORROPIA. For Branch 3, each of TS, TA and TN > tiny, as well as one (or both) of TNa and TCl. For Branch 4, each of TS, TA and TN > tiny, as well as one (or all) of TMg, TK and TMg – thus Branch 4 does not necessarily require TNa or TCl > tiny. However, it should be noted that for a solution to be possible, subcases CALCH6, CALCG5, CALCM8, CALCO7 and CALCP13 do require TCl > tiny. The dashed lines in the figure implies that the domain extends infinitely in the direction of increasing R1 or R2. For example, in Branch 1, $0 \leq R_1 < \infty$, but in the figure $R_1$ only extends to 4, and subcase CALCA2 extends for all TA/TS >2.

## 3 Algorithm design and improvements

During the development of HETP, several improvements related to the mathematical techniques were incorporated relative to ISORROPIA (and HETV), as well as additional modifications related to mass balance. These modifications and improvements include:

(1) An updated root finding algorithm, referred to as 'interpolate, truncate and project (ITP)' (Oliveria et al., 2021), has been used instead of the bisection method in HETP. ITP has the advantage of 'superlinear convergence', and hence may obtain a root with the same accuracy as bisection, but in less iterations. The increased rate of convergence can affect the activity coefficients. In some cases, the faster convergence of ITP can alter the ionic strength, resulting in different activity coefficients being calculated early on in the iterative process than would be determined from the

bisection algorithm used in ISORROPIA. The new approach may also contribute to an improved formal accuracy performance for estimating the roots, for the same convergence criteria level (see Sect. 4.1).

(2) All bisection subroutines in ISORROPIA employ a root bracketing approach to obtain an initial interval $[x_a, x_b]$ where $f(x_a)f(x_b) < 0$, signifying that a root exists within the interval according to the intermediate value theorem, assuming a continuous function. We have found that ISORROPIA does not check to determine if either endpoint is a valid root, that is, if $f(x_a) = 0$ or $f(x_b) = 0$. Instead, ISORROPIA will proceed to the next interval, continuing its search for a root and potentially locating a different root than expected (the code seeks the smallest positive real root in the case of multiple roots in the search domain), or a slower convergence towards the start or end of the root interval than might otherwise be the case. In HETP we have included a check during the root bracketing stage to identify cases when $x_a$ or $x_b$ is a valid root. If an endpoint is a root, then HETP will return since an equilibrium solution has been found. It should be noted that the occurrence of an endpoint as a valid root is extremely rare and hence neglecting this modification will have no effect on most output from the solver, but nonetheless we have included this possibility in HETP for completeness and accuracy.

(3) In some cases that require ITP (or bisection in ISORROPIA) to obtain an equilibrium solution, the independent variable (i.e., $x$) converges, but the function being evaluated at $x$ (i.e., $y = f(x)$) oscillates between a negative and positive value, and thus $|y|$ does not converge to zero as expected if $x$ is a root (despite convergence of $x$). This oscillating behavior of $y$ may indicate (i) that $x$ is a discontinuity, (ii) that there is significant non-linearity in the partitioning solution, or (iii) that the accepted tolerance on $x$ is too loose for convergence, and hence $x$ is not an accurate solution to the system of equations at the targeted tolerance level for $x$. For all subroutines requiring ITP, HETP will track the species concentrations, activity coefficients and the value of $x$ that are found to minimize $|y|$ during the iterative process. If after convergence of $x$ it is determined that $|y|$ is not minimized compared to all earlier iterations, then HETP will 'reset', and instead use the $x$ value, species concentrations and activity coefficients that were found to minimize $|y|$ – this is chosen as the solution of the system. The effect of this modification on the output from HETP is discussed in Sect 4.2.

(4) In all chemical subspaces, a quadratic equation must be solved for a subsystem of the equations, while in some cases a cubic equation will be solved. Quadratic equations have the form $f(x) = ax^2 + bx + c$, where the solution corresponding to $f(x) = 0$, is usually expressed as the standard quadratic formula $x = \frac{-b \pm \sqrt{b^2 - 4ac}}{2a}$. $x$ has two possible solutions, $x_1$ and $x_2$, determined by the sign in front of the radical. As identified in Makar et al., (2003) in the original version of HETV, when the coefficient '$b$' differs by several orders of magnitude from coefficients '$a$' or '$c$', floating-point arithmetic can fail to give an accurate answer for $x$ when using the standard root formula. For example, if $\sqrt{b^2 - 4ac} \approx b$, then addition in the quadratic formula may be problematic since we are subtracting two nearly equal numbers (i.e., $\approx -b + b$). To avoid this issue, HETP uses the analytic formula given in Press et al.,

(2007) to solve the quadratic equation: $q = -\frac{1}{2}\left(b + sign(b)\sqrt{b^2 - 4ac}\right)$ with roots $x_{p_1} = \frac{c}{q}$ and $x_{p_2} = \frac{q}{a}$. Care must be taken when applying this formula since the appropriate choice of $x_{p_1}$ and $x_{p_2}$ depends simultaneously on the chosen solution (i.e., $x_1$ or $x_2$) and the sign of the $b$ coefficient, as described in Table S3 of the supplement. In addition to the analytic formula from Press et al., (2007), HETP also includes code (which is commented out) to solve the quadratic equation using a Taylor series expansion of the quadratic formula. In this code, the Taylor series expansion is only applied when the coefficients '$b$' and '$c$' differ by orders of magnitude, and hence when the numerical precision issues as described above are likely to occur (note that $a = 1$ in all subroutines; formulae were normalized). Both methods produce very similar results, but the analytic formula provided by Press et al., (2007) is superior to the Taylor expansion since it provides an exact solution, giving lower error metrics (i.e., Sect. 4). For cases where a cubic equation must be solved, HETP will employ an ITP search to obtain an estimate of the smallest positive real root if an exact analytic solution is not possible. The generic formulae describing the exact analytic solution of a cubic polynomial is from Spiegel et al., (2009) and is used in ISORROPIA. It should be noted that the requirement to solve a cubic equation occurs only during the solution procedure of the minor systems of I6, J3, L9 and K4. For example, the call to solve a cubic equation occurs on line 130 of subroutine 'mach_hetp_calchclhno3'. The most recent version of ISORROPIA (i.e., ISORROPIA-lite) did not address these outstanding numerical issues.

(5) During the development of HETP we have identified several cases where a negative ion or gas concentration can be output from ISORROPIA. For example, a negative concentration of $NH_4^+$ can occur when solving the minor system $NH_{3(g)} + H^+_{(aq)} \leftrightarrow NH_4^+_{(aq)}$ for thermodynamic equilibrium. In this case, HETP and ISORROPIA will solve a quadratic equation to determine the concentration of ammonia gas ($NH_3$). From the concentration of $NH_3$, the ammonium cation is determined as $NH_4^+ = NH_4^+_i - NH_3$, where $NH_4^+_i$ is the ammonium concentration determined from the *major* system (see Table S2, Supplemental Information). If partitioning (after solving the quadratic equation) at this stage gives $NH_3 \approx NH_4^+_i$, then subtraction of two nearly identical numbers may lead to a floating-point arithmetic error and a final concentration of $NH_4^+ < 0$ in the original ISORROPIA equations. In HETP, negative output is strictly prohibited. To accomplish this, we have utilized max statements that force any negative concentrations to zero, in conjunction with the more accurate evaluation of the quadratic formula (i.e., point 4 above).

(6) In ISORROPIA, the initial dry salt partitioning that is completed at the commencement of chemical subspace L9 may fail to conserve mass for sulfate, ammonium, potassium, and sodium, in some cases. In HETP we have slightly modified the initial dry salt partitioning of CALCL9 (see Table S2, Supplemental Information) to ensure mass conservation holds for all cases. Any free TA that may result in L9 is assumed to be in the gas phase as $NH_3$ and is added back to the final equilibrium solution after convergence of both the major and minor systems. As discussed in Sect. 2, the free amounts of $SO_4$, Na, Mg, K and Ca are explicitly tracked in HETP for all chemical subspaces and returned to the calling code to prevent a loss of mass in the output speciation.

(7) Mass conservation may not hold in ISORROPIA when the input precursor concentrations are near the lower limit for species concentrations, "tiny" ($1\times10^{-20}$ mol m$^{-3}$), used in the solver. The same lower limit used to bound the input precursor concentrations is also used throughout ISORROPIA to bound the species concentrations during and after chemical partitioning. In HETP we use the same lower limit as ISORROPIA to bound the input precursor species (i.e., tiny), but during and after partitioning the lower limit for gaseous speciation is reduced to tiny2 = $1\times10^{-28}$ mol m$^{-3}$. This reduction of the lower limit for gaseous speciation during the iterative process improves mass conservation for the limiting case when the input precursor concentrations are near the lower limit of tiny.

(8) The subroutine 'adjust' performs a post-convergence mass balance adjustment for ammonium, sulfate, nitrate, and chloride, with the goal of ensuring mass conservation holds to machine precision. Specifically, this subroutine checks only for *excess* mass relative to the input totals. If identified, the excess mass is removed first from the aqueous phase, and then from the solid phase, and finally from the gaseous phase, until no excess remains. However, the mass adjustment of sulfate in ISORROPIA does not include CaSO$_4$ in the mass balance calculations, and therefore in some cases, ISORROPIA will fail to properly conserve mass to machine precision. In HETP we have included CaSO$_4$ in the mass balance adjustment of sulfate.

(9) Improvements to the overall code structure and efficiency include:

    (a) Use of modern Fortran compared to FORTRAN 77 in ISORROPIA,

    (b) Use of explicit declarations only – all subroutines now start with an 'implicit none' statement and all common blocks have been removed,

    (c) Removing all GOTO statements, and instead using modern Fortran constructs such as 'do while' loops,

    (d) Removing function and short subroutine calls, except for process calls to calculate activity coefficients (calcact), to solve a cubic equation (poly3), to solve minor systems, and to perform a post-convergence mass balance adjustment (adjust). The merging of functions and some short subroutines allowed several variables to be calculated once and reused throughout the iterative process, reducing computational time,

    (e) Moving expressions being recalculated unnecessarily within loops to take place prior to the loop, and removing calculations that serve no purpose to the actual solution being sought,

    (f) Pre-calculating constant values which are then stored as variables to be used later in the subroutine and,

    (g) Designing the code to include an optional use of a vectorization-by-grid point approach (Makar, 1995), which may reduce the call factor overhead on some compilers.

## 4 Comparison between HETP and ISORROPIA

### 4.1 Case-by-case comparison

In this section the output from HETP is compared to ISORROPIA for a set of 10,000 artificially generated input 'test cases' that span the domain of each chemical subspace. The test cases have all precursor species held constant except the total sulfate

(TS) which is slowly varied linearly over the range of the chemical subspace. Tests of this nature demonstrate the stability of numerical solutions – adjacent tests along the same axis of variation in general are expected to be smoothly varying (Makar et

al., 2003). The convergence criteria are consistent between both solvers. For activity coefficients, $\epsilon_{act} = 1\times10^{-6}$ and $maxit_{act}$ = 4, where $\epsilon_{act}$ is the relative error limit between successive iterations of activity coefficient calculations, and $maxit_{act}$ is the maximum number of allowed iterations. For bisection or ITP, $\varepsilon = 1\times10^{-9}$, $maxit_{bsec} = 100$ and $ndiv = 5$, where $\varepsilon$ is defined in Sect. 1, $maxit_{bsec}$ is the maximum number of allowed iterations, and $ndiv$ is the number of subdivisions searched for an interval containing a root (i.e., sign change) prior to the start of bisection or ITP. All output from HETP, in this section and

those presented hereafter, includes the modifications outlined in Sect. 3 unless stated otherwise. The ISORROPIA code used in this comparison is the base version (ISORROPIA v2.2) used in the CMAQ air-quality model (USEPA, 2022). ISORROPIA throughout this paper has been compiled using the '-r8' flag that converts all real variables to double precision, to ensure the precision of both solvers is consistent (HETP uses double precision throughout). It should be noted that ISORROPIA is coded to use mostly 'double precision' variables, but some single precision variables exist (i.e., declared as 'real', either explicitly or

by default under Fortran variable naming conventions). While compiling ISORROPIA with the intel Fortran compiler flag '-r8' does not have a large impact on the execution time, it may in some cases produce non-trivial differences in the output, compared to output produced without the '-r8' flag. Aside from the '-r8' flag, no other compilation flags were used in this work. All numerical tests herein were executed on a Lenovo ThinkSystem SD650v2 DWC computer, which uses an Intel® Xeon® Platinum 8380 CPU running at a clock speed of 2.30 GHz, with 512 GB of available random-access memory. The

compiler used was an intel compiler (IFORT) version 2021.5.0.2021109.

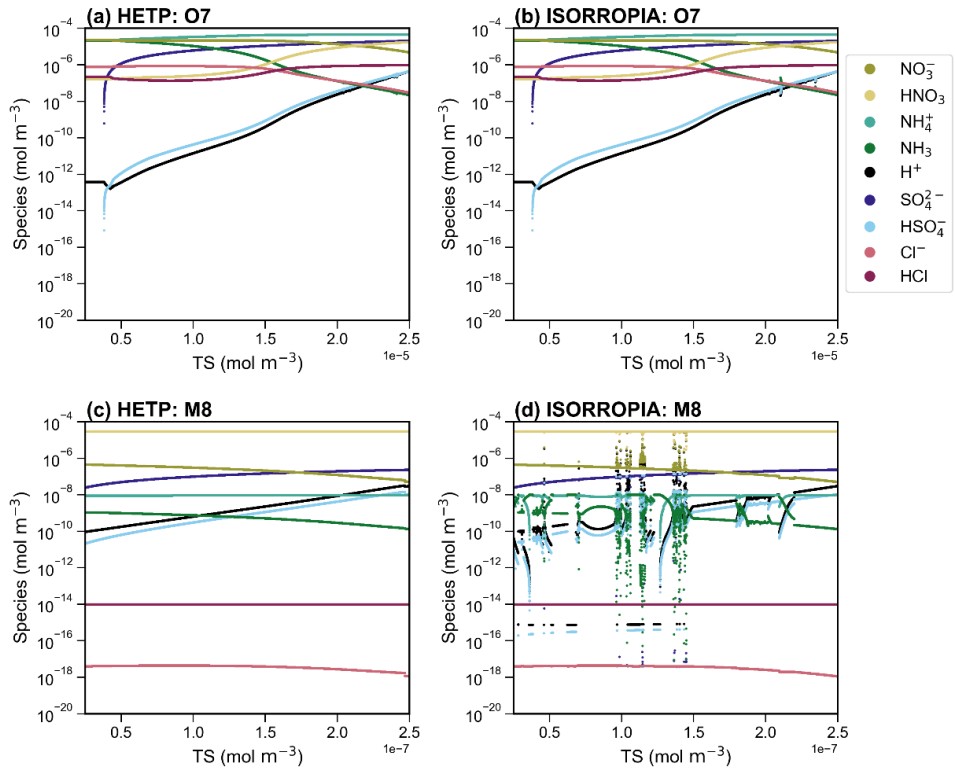

**Figure 2:** A side-by-side comparison of the output from HETP (left) and ISORROPIA (right), for the chemical subspace CALCO7 (a-b) and CALCM8 (c-d). All input species are held constant, except the total available sulfate (TS) which is varied over 10,000 sets of initial conditions. The air temperature and relative humidity are 306 K and 35 % respectively, for all test cases in the figure. The convergence criteria are consistent between both solvers (see text).

Figure 2 displays the output from ISORROPIA and HETP for two example chemical subspaces: (a-b) displays CALCO7 and (c-d) shows CALCM8. These chemical subspaces involve the presence of at least one of $Ca^{2+}$, $K^+$ and $Mg^{2+}$ and so they were not included in the original HETV package, which was designed for the $SO_4$–$NO_3$–$NH_4$–$H_2O$ system. Furthermore, these two subspaces are frequently called in practical CTM applications (see Sect. 4.2) and hence are used to compare HETP against ISORROPIA in this section. For the test cases shown in Fig. 2, the relative humidity (RH) was set to 35 % and the air temperature ($T$) to 306 K, conditions typical of a hot summer day in central North America. The output for CALCO7 is nearly identical between the two solvers with a difference of $< 1$ % between HETP (Fig. 2a) and ISORROPIA (Fig. 2b), except for TS between $2.1\times10^{-5}$ and $2.4\times10^{-5}$ mol m$^{-3}$, where visual differences begin to appear, particularly for $H^+$, $HSO_4^-$ and $NH_3$. In the case of CALCM8, the output from HETP (Fig. 2c) is vastly different from ISORROPIA (Fig. 2d) for the same initial conditions and convergence criteria. For these initial conditions, the ISORROPIA solution shows the effects of numerical instability in the bisection root-finding procedure. The ISORROPIA algorithm used in CALCM8 is designed so

that the variable being bisected is proportional to Cl⁻ (see Table S2, Supplemental Information). At the same time, the multicomponent activity coefficients are dependent on the ionic strength of the aqueous aerosol, determined from the molar concentration of all ions present, including Cl⁻. Both of these iterative procedures are completed simultaneously, and impact each other in a nonlinear fashion. The choice of Cl⁻ during the first iteration of bisection (or ITP) may considerably impact the final equilibrium solution, by altering the initial ionic strength, and as a result, the convergence of the multicomponent activity coefficients. This effect is demonstrated for CALCM8 in Fig. 2(c-d), where the differences between ISORROPIA and HETP are related only to the choice of root-finding methodology. In fact, if the ITP approach within HETP is reverted to the same bisection algorithm used in ISORROPIA, then the output from HETP begins to show the same unstable behaviour that is demonstrated in the ISORROPIA simulation shown in Fig. 2d. It should be noted that these differences are due to the choice of root-finding methodology and are not the result of allowing the ends of the interval to be potentially valid roots (i.e., point 2 in Sect. 3).

The accuracy of each solver can be assessed directly by introducing an error term ($\xi$), determined as the logarithmic difference between the 'calculated' equilibrium constant ($K_{calc}$) and the 'true' equilibrium constant ($K_{true}$), that is, $\xi = \log(K_{calc}) - \log(K_{true})$. $K_{calc}$ is determined from the species concentrations converted to molalities using the aerosol liquid water content in kg m⁻³ and activity coefficients after convergence of the major or minor system (i.e., from the equations in Table 1), while $K_{true}$ is calculated from the Van't Hoff equation (Eq. 1). The parameter $\xi$ thus provides a direct measure of each solver's proximity to the actual root of the system of equations, for a given level of convergence criteria employed in both solvers. For statistical characterization of $\xi$, the absolute value of the difference is used, so that $\xi' = |\xi|$. A logarithmic difference is used herein (instead of a percent difference, for example) since the difference between $K_{calc}$ and $K_{true}$ can span several orders of magnitude. In this way, a difference on the order of 1 implies that $K_{calc}$ and $K_{true}$ differ by an order of magnitude, while a difference on the order of $1 \times 10^{-2}$ implies $K_{calc}$ and $K_{true}$ differ starting at the second or third digit, when written in scientific notation. The error analysis has been completed using the case-by-case implementation of HETP (see Sect. 4.3). Ideally, $\xi' = 0$, signifying that the problem has converged to a solution whose concentrations and activity coefficients satisfy the equilibrium equations of the major and minor systems precisely. In reality, however, there may be some magnitude of difference between $K_{calc}$ and $K_{true}$. The accuracy of $K_{true}$ calculated from Eq. 1 (used in both solvers) is limited to 3 significant digits due to the variable $-\Delta H_f^0/(RT_0)$. Therefore when $\xi' < 1 \times 10^{-3}$ in either solver, we can conclude that $K_{calc}$ after convergence is identical to $K_{true}$ within its known accuracy. However, in practical applications (i.e., within a CTM), the value of $K_{true}$ calculated from Eq. 1 will retain all digits as determined by the precision of the code (i.e., double precision in HETP) and therefore $\xi'$ may be $\ll 1 \times 10^{-3}$. Hence, we seek a solver that obtains $\xi'$ as close to zero as possible. Table 2 gives the median, the maximum, and the 25th and 75th percentiles of $\xi'$ for HETP and ISORROPIA, corresponding to the data presented in Fig. 2. For CALCM8, the median $\xi'$ is lower in HETP than ISORROPIA for all equilibrium constants, which suggests that HETP is obtaining a more accurate solution for this set of input conditions. The difference in median $\xi'$ between the two solvers is large and indicates that HETP values are more accurate than ISORROPIA by many orders of

magnitude, for the same level of convergence criteria. For example, for $K_{HCl}$ HETP has a median $\xi' \approx 1.77 \times 10^{-8}$, while ISORROPIA has a median $\xi' \approx 0.39$, with similar results for $K_{HNO_3}$. The superior performance of HETP for this set of initial conditions can also be confirmed visually by comparing Fig. 2(c) to Fig. 2(d). For all species present in this subspace, HETP shows a smooth transition with incremental change in TS, but this is not the case for ISORROPIA. In CALCM8, the very large differences in median $\xi'$ between the two codes demonstrates that the $\xi'$ values are linked to the poor convergence performance of ISORROPIA and are associated with the high degree of sensitivity of that algorithm's use of bisection towards initial conditions.

In CALCO7 (Fig. 2a-b), the median $\xi'$ for all equilibrium constants is lower in HETP than ISORROPIA, but the difference between the two solvers is marginal, especially when the 25th and 75th percentiles are considered (i.e., for $K_{HCl}$ the 75th percentile of $\xi'$ is $4.88 \times 10^{-7}$ and $4.40 \times 10^{-7}$ for HETP and ISORROPIA respectively). Table 2 also gives statistics of $\xi'$ for the same set of input precursor concentrations, but now with a RH = 65 % and $T$ = 263 K. The main difference here is that CALCO7 performs slightly worse in HETP than ISORROPIA, as determined from the median and 75th percentile of $\xi'$. Despite this worse statistical performance in HETP, there are no visual differences between them when the output from each solver is plotted (see Fig. S1). In this case, the median $\xi'$ of both solvers is on the order of $1.0 \times 10^{-4}$, implying that the difference between $K_{calc}$ and $K_{true}$ occurs in the 4th digit when written in scientific notation. As a result, the differences between HETP and ISORROPIA do not become apparent unless the graph is zoomed in very close to the data points. For CALCM8 at these new meteorological conditions (RH = 65 % and $T$ = 263 K), HETP has an unstable behavior in the output speciation for TS between $1.6 \times 10^{-7}$ and $2.3 \times 10^{-7}$ mol m$^{-3}$, while ISORROPIA has an unstable behavior for all TS > $0.7 \times 10^{-7}$ mol m$^{-3}$ (see Fig. S1). This poor performance in CALCM8 for these meteorological conditions is demonstrated in the statistics of $\xi'$ shown in Table 2.

**Table 2:** Theoretical error ($\xi'$) for $n = 10{,}000$ generated input conditions corresponding to the chemical subspaces O7, M8 and I6. Statistics of $\xi'$ for two sets of atmospheric conditions are presented (temperature, $T$ and relative humidity, RH). The bolded values denote the smallest median error for that equilibrium constant (i.e., row) between HETP and ISORROPIA.

| Case | Equilibrium Constant | HETP: $\xi' = |\log(K_{\text{true}}/K_{\text{calc}})|$ | | | | ISORROPIA II: $\xi' = |\log(K_{\text{true}}/K_{\text{calc}})|$ | | | |
|---|---|---|---|---|---|---|---|---|---|
| | | **Median** | **Q25** | **Q75** | **Maximum** | **Median** | **Q25** | **Q75** | **Maximum** |
| *T* = 306 K; RH = 35 % (Fig. 2) | | | | | | | | | |
| **O7** | $K_{\text{NH}_3}/K_{\text{H}_2\text{O}}$ | **9.83×10⁻⁹** | 3.56×10⁻¹² | 2.09×10⁻⁶ | 1.81×10⁻³ | 9.84×10⁻⁹ | 3.51×10⁻¹² | 2.09×10⁻⁶ | 1.81×10⁻³ |
| | $K_{\text{HNO}_3}$ | **1.27×10⁻⁹** | 2.63×10⁻¹⁰ | 4.88×10⁻⁷ | 0.30 | 2.86×10⁻⁹ | 1.19×10⁻⁹ | 4.80×10⁻⁷ | 0.85 |
| | $K_{\text{HCl}}$ | **1.27×10⁻⁹** | 2.63×10⁻¹⁰ | 4.88×10⁻⁷ | 0.30 | 2.86×10⁻⁹ | 1.19×10⁻⁹ | 4.40×10⁻⁷ | 0.85 |
| **M8** | $K_{\text{NH}_3}$ | **2.52×10⁻¹³** | 2.13×10⁻¹⁴ | 3.61×10⁻¹² | 6.95×10⁻¹¹ | 7.33×10⁻¹² | 3.55×10⁻¹⁴ | 1.47×10⁻¹⁰ | 12.0 |
| | $K_{\text{HNO}_3}$ | **1.77×10⁻⁸** | 1.89×10⁻⁹ | 9.34×10⁻⁸ | 1.87×10⁻⁴ | 0.39 | 8.38×10⁻⁸ | 1.93 | 40.5 |
| | $K_{\text{HCl}}$ | **1.77×10⁻⁸** | 1.89×10⁻⁹ | 9.34×10⁻⁸ | 1.87×10⁻⁴ | 0.39 | 8.38×10⁻⁸ | 1.94 | 30.8 |
| *T* = 263 K; RH = 65 % (Fig. S1) | | | | | | | | | |
| **O7** | $K_{\text{NH}_3}/K_{\text{H}_2\text{O}}$ | 2.31×10⁻⁵ | 1.00×10⁻⁹ | 2.49×10⁻³ | 5.19×10⁻² | 2.31×10⁻⁵ | 9.69×10⁻¹⁰ | 2.49×10⁻³ | 5.19×10⁻² |
| | $K_{\text{HNO}_3}$ | 3.60×10⁻⁴ | 4.70×10⁻¹⁰ | 2.95×10⁻³ | 7.69×10⁻³ | **1.52×10⁻⁴** | 9.13×10⁻¹⁰ | 9.10×10⁻⁴ | 3.19×10⁻³ |
| | $K_{\text{HCl}}$ | 3.60×10⁻⁴ | 4.70×10⁻¹⁰ | 2.95×10⁻³ | 7.69×10⁻³ | **1.52×10⁻⁴** | 9.13×10⁻¹⁰ | 9.10×10⁻⁴ | 3.19×10⁻³ |
| **M8** | $K_{\text{NH}_3}$ | **1.84×10⁻¹¹** | 7.38×10⁻¹² | 3.59×10⁻¹¹ | 31.8 | 2.32 | 8.87×10⁻¹¹ | 11.5 | 17.1 |
| | $K_{\text{HNO}_3}$ | **1.75** | 1.67 | 1.90 | 7.17 | 3.89 | 1.96 | 20.3 | 37.4 |
| | $K_{\text{HCl}}$ | **1.74** | 1.67 | 1.90 | 7.17 | 3.92 | 1.95 | 20.9 | 25.8 |
| *T* = 243 K; RH = 5 % (Fig. 4) | I6–1: No improvements to root–finding methodology in HETP  I6–2: Updated analytic formula to solve quadratic equations, no ITP for cubic equations  I6–3: Updated analytic formula to solve quadratic equations and ITP for cubic equations | | | | | | | | |
| **I6–1** | $K_{\text{HSO}_4}$ | 9.02 | 1.68 | 15.1 | 35.9 | 9.80 | 3.87 | 18.1 | 40.6 |
| | $K_{\text{HNO}_3}$ | 15.1 | 11.7 | 18.3 | 24.8 | 15.1 | 11.9 | 18.2 | 23.8 |
| | $K_{\text{HCl}}$ | 15.1 | 11.7 | 18.3 | 24.8 | 15.1 | 11.9 | 18.2 | 23.8 |
| **I6–2** | $K_{\text{HSO}_4}$ | 2.94 ×10⁻⁵ | 6.67 ×10⁻⁸ | 3.66 ×10⁻² | 5.75 | — | — | — | — |
| | $K_{\text{HNO}_3}$ | 13.0 | 9.35 | 16.2 | 19.8 | — | — | — | — |
| | $K_{\text{HCl}}$ | 13.0 | 9.35 | 16.2 | 19.8 | — | — | — | — |
| **I6–3** | $K_{\text{HSO}_4}$ | 2.94 ×10⁻⁵ | 6.67 ×10⁻⁸ | 3.66 ×10⁻² | 5.75 | — | — | — | — |
| | $K_{\text{HNO}_3}$ | 1.46×10⁻⁹ | 6.38×10⁻¹⁰ | 3.36×10⁻⁸ | 1.45×10⁻² | — | — | — | — |
| | $K_{\text{HCl}}$ | 1.46×10⁻⁹ | 6.38×10⁻¹⁰ | 3.36×10⁻⁸ | 1.45×10⁻² | — | — | — | — |

Figure 3 displays a comparison of HETP and ISORROPIA, where now TS and TA are varied simultaneously while all other input precursor species are held constant. Figure 3 displays output generated from $n = 2{,}000{,}000$ unique test cases. These test cases are divided into two tests, denoted as a high $Mg^{2+}$-$Ca^{2+}$-$K^+$-$Na^+$ case (Fig. 3a-c) and a low $Mg^{2+}$-$Ca^{2+}$-$K^+$-$Na^+$ case (Fig. 3d-f). The input conditions used to generate Fig. 3 are summarized in Table 3. It should be noted that in the unaltered version of ISORROPIA, TCl < $1\times10^{-14}$ mol m⁻³ would have necessitated a mass adjustment at the commencement of the solver. In this case, TCl would be reset to a floor value of $1\times10^{-10}$ mol m⁻³, thereby creating mass. This adjustment has not been applied here.

**Table 3** – Input conditions used to generate Fig. 3 for 2,000,000 total unique test cases. All input precursor species have units of mol m$^{-3}$.

| | Fig(a-c): High Mg$^{2+}$-Ca$^{2+}$-K$^+$-Na$^+$ | Fig(d-f): Low Mg$^{2+}$-Ca$^{2+}$-K$^+$-Na$^+$ |
|---|---|---|
| **TS** | Varying between $2.5\times10^{-5}$ and $2.5\times10^{-12}$ | Varying between $2.5\times10^{-5}$ and $2.5\times10^{-12}$ |
| **TA** | Varying between $2.5\times10^{-5}$ and $2.5\times10^{-18}$ | Varying between $2.5\times10^{-5}$ and $2.5\times10^{-9}$ |
| **TN** | $3.0\times10^{-6}$ | $1\times10^{-8}$ |
| **TNa** | $1.0\times10^{-5}$ | $1\times10^{-6}$ |
| **TCl** | $1.0\times10^{-14}$ | $1\times10^{-14}$ |
| **TCa** | $1.0\times10^{-8}$ | $1\times10^{-16}$ |
| **TK** | $1.0\times10^{-14}$ | $1\times10^{-17}$ |
| **TMg** | $1.0\times10^{-14}$ | $1\times10^{-16}$ |
| **Temp (K)** | 306 | Same as Fig(a-c) |
| **RH (%)** | 35 | Same as Fig(a-c) |
| **$n$** | 1,000,000 | 1,000,000 |

The colors in Fig. 3(a-b) and Fig. 3(d-e) represent the amount of gaseous NH$_3$ after partitioning between the gas and aerosol phase. The test input spans across all of Branch 4 (O7, M8, P13, L9 and K4), using the same convergence criteria as Fig. 1 and Fig. 2. The colors shown in Fig. 3c and Fig. 3f give the absolute percent difference between Fig. 3a and Fig. 3b, and Fig. 3d and Fig. 3e, respectively, calculated relative to HETP as |HETP-ISO|/HETP×100 %. The color contour intervals in Fig. 3 are on a logarithmic scale. In each figure panel dashed black lines separate between the different chemical subspaces, with the particular subspace label superimposed. In Fig. 3(a-b) and Fig. 3(d–e) the output compares well between HETP and ISORROPIA for the subspaces O7, M8 and P13, with absolute differences typically $< 0.1$ % and no obvious visual differences between the two solvers. However, in the high Mg$^{2+}$-Ca$^{2+}$-K$^+$-Na$^+$ case (Fig. 3a-b), there are some noticeable visual differences between the two solvers for the subspaces K4 and particularly L9. The differences in L9 between the two solvers result from (i) the updated methodology within HETP to calculate polynomial roots, (ii) a correction within HETP to the initial dry salt partitioning to ensure mass conservation, and (iii) one less call to calculate activity coefficients in HETP for some test cases, specifically those test cases that have no convergence of activity coefficients after completing the maximum number of allowed iterations. The largest absolute differences of 100 to 600 % are in L9, and are predominantly due to (ii), where for some input conditions ISORROPIA creates dry salt mass for TA, TS, and TK. Specifically in ISORROPIA, 6.02%, 0.05% and 5.97% of the test input conditions shown in Fig. 3(a-b) create mass for TS, TA, and TK respectively that cannot be attributed to machine precision near the lower limit used in the solver (i.e., Species$_{out}$ − Species$_{in}$ $> 9.999\times10^{-19}$ mol m$^{-3}$). The median relative mass created for these input conditions is 22.6% for TS, 0.24% for TA and $2.93\times10^{10}$ % for TK. In K4, (ii) is not applicable, so the differences are thus due to (i) and (iii). As demonstrated in Fig 3(a-c), there is a large amount of 'noise' in K4 for TS $> 1\times10^{-5}$ mol m$^{-3}$ and TA $< 12\times10^{-6}$ mol m$^{-3}$ in ISORROPIA that is not present in HETP. This 'noise' shows up as speckling in the percent difference plots and is due mainly to (i). If the noise in ISORROPIA is neglected for K4, then the output from ISORROPIA is quite similar to HETP, with differences $< 1\%$.

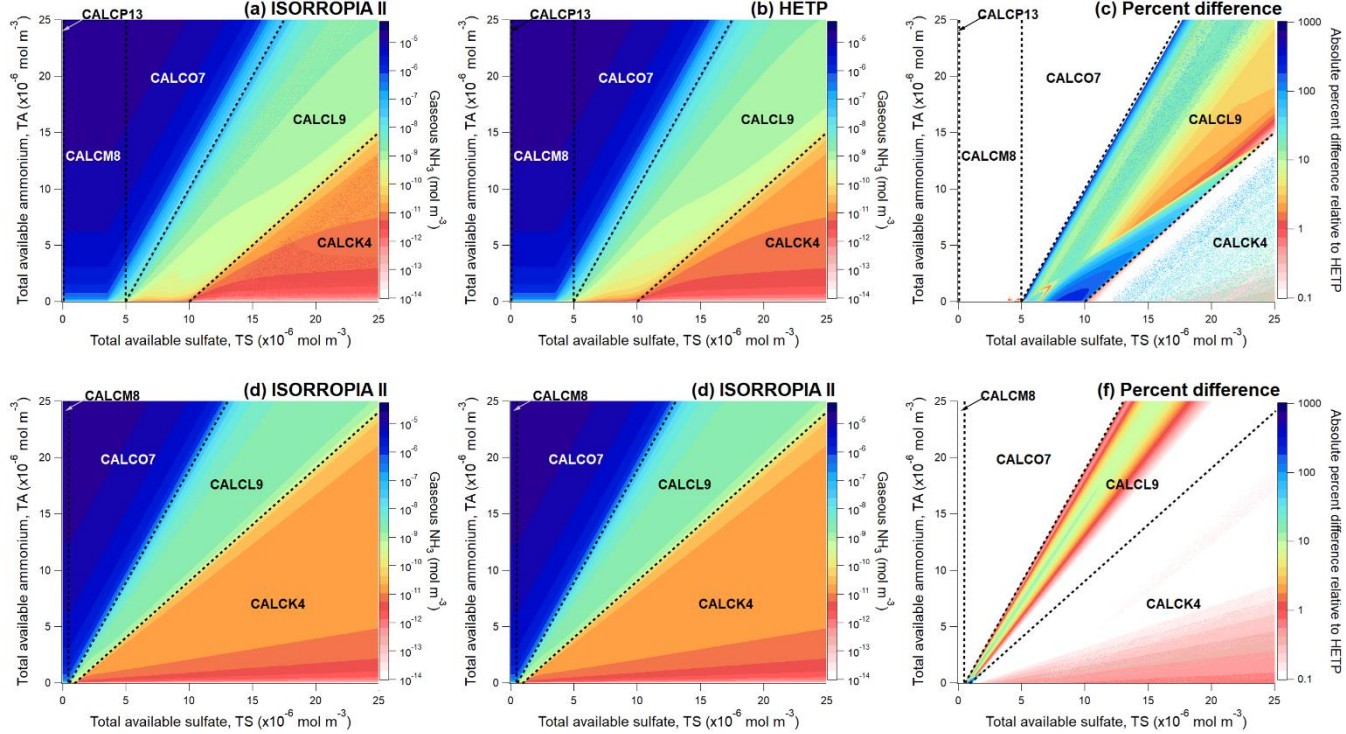

**Figure 3:** Regular variation (linear) of the total available sulfate (TS) and the total available ammonium (TA), while holding all other input precursor species constant (see Table 3 for a summary of the input conditions). The colors in figure panels (a-b) and (d-e) give the amount of gaseous $NH_3$ after chemical partitioning at thermodynamic equilibrium (the color scale is logarithmic and identical in these panels). Panel (c) and (f) show the percent difference of (a-b) and (d-e) respectively. Each panel set (i.e., a-c and d-f) includes 1,000,000 unique input test cases, with the same convergence criteria as used to generate Fig. 2. Superimposed on each panel are dashed black lines denoting the boundaries between different chemical subspaces. The actual subspace contained within a set of dashed lines is given as a text label.

An additional concern identified in Makar et al., (2003) is the potential impact of the inaccurate evaluation of the quadratic and cubic formula (i.e., analytic formulae to obtain an 'exact' solution), which remains present in subsequent iterations of ISORROPIA since the development of HETV (see Sect 3, point 4). An example showing the incremental improvement of the quadratic and cubic solution procedure on the output speciation is displayed in Fig. 4, which depicts the output of CALCI6 from ISORROPIA (Fig. 4b) and HETP (Fig. 4a, c-d). This case illustrates differences that would occur at rather low temperatures and relative humidity, in this case $T$ = 243 K and RH = 5 %. While such a combination of air temperature and relative humidity is likely to be rare in the lower troposphere, it is not uncommon for surface air temperatures to reach 243 K or lower in the winter in Canada, and at similar higher latitudes in other parts of the world. The choice of RH = 5 % here is used to highlight the numerical issues present in ISORROPIA, which occur more frequently and are more pronounced at low RH. However, the numerical issues highlighted in Fig. 4 continue to be present in the output from ISORROPIA, but to a lesser extent, even as the RH is increased to 35 % for the same set of initial conditions. At an ambient RH of 5 % the assumption of a supersaturated aqueous phase may be less justified and is more likely to be representative of a

very hypothetical case.  Nonetheless, observations from southern California have indicated (although in warmer air temperatures than investigated here) that crystallization of some ambient aerosols may not occur until a RH as low as 4 % (Shaw and Rood, 1990), suggesting that in some atmospheric conditions metastable aerosols are possible even at a very low RH of 5 %.

In Fig. 4a, HETP has been executed without any modifications to improve the accuracy of polynomial root
calculations, so that the only improvement over ISORROPIA is that HETP will not allow negative species concentrations (i.e., $HSO_4^-$).  In Fig. 4c, HETP now includes an improved methodology to calculate roots of quadratic polynomials ('analytic quad'), in addition to the improvement related to negative species concentrations of Fig. 4a.  Lastly, In Fig. 4d, HETP now includes an ITP search to determine the roots of cubic polynomials, in addition to the improvements of Fig. 4a and c.  Figure 4 follows the same procedure as Fig. 2 – that is, an incremental variation of the input TS while holding all other precursor
species constant.  Without the modifications applied in Fig. 4c and d, the output from HETP and ISORROPIA are quite similar.  However, as numerical improvements are incrementally applied to HETP, clear visual differences between HETP and ISORROPIA become apparent for most chemical species in this subspace.  In CALCI6, the major system being solved is $H^+$– $HSO_4^-$–$SO_4^{2-}$, requiring a quadratic root with a large variation in coefficient magnitudes to be derived – and therefore an error in $H^+$ will propagate through to the minor systems that are solved thereafter (see Table S2, Supplemental Information).  It
should be noted that the $y$-axis in Fig. 4 is logarithmic, so negative values are not shown in the figure panels.  Nonetheless, there are many instances when ISORROPIA outputs a negative concentration of $HSO_4^-$ for this subspace (Fig. 4b) as a result of the use of the standard (and under these circumstances inaccurate) formula for the roots of a quadratic equation for $H^+$ in this subspace.  In HETP updated methodologies are used to solve the quadratic equation, that avoid numerical inaccuracies due to catastrophic cancellation.  The result of this modification (as demonstrated in Fig. 4c) is the removal of the numerical
instability present in the output of HETP for this set of initial conditions shown in Fig. 4a.  Numerical instability caused by the erroneous evaluation of the quadratic formula appears to be most prevalent at a low relative humidity (low aerosol water mass).

Following convergence of the major system in CALCI6, the minor systems are solved, one of which requires the roots of a cubic polynomial to be identified; the smallest positive real root determines the concentration of $Cl^-$ and $NO_3^-$.  In HETP, an ITP search is employed to determine the smallest positive real root of the cubic polynomial when an exact analytic
solution from the cubic root formulae is not possible, due to a large range in the magnitude of the coefficients of the cubic polynomial, which may lead to floating-point arithmetic errors.  For the set of input conditions shown in Fig. 4, including an ITP search to solve cubic polynomials results in about 72 % more roots being identified in HETP than in ISORROPIA.  If ISORROPIA is unable to determine a valid root from the cubic formula, it will assume that the root is a tiny value (i.e., $1\times10^{-20}$ mol m$^{-3}$) – this is the procedure that was applied to generate the output shown in Fig. 4a-c.  The effect of including an ITP
search to solve cubic polynomials is a very large reduction in $\xi'$ for $K_{HCl}$ and $K_{HNO_3}$ in the chemical subspaces I6, J3, L9 and K4 for some sets of initial conditions.  Statistics of $\xi'$ corresponding to CALCI6 shown in Fig. 4 are given at the bottom of Table 2.  For example, in Fig. 4d, HETP has been implemented with an ITP search to solve cubic polynomials, and as shown in Table 2, this implementation leads to a large reduction in the median $\xi'$ for $K_{HCl}$ from 13.0 to $1.46\times10^{-9}$.  The difference

here is a solution that is accurate versus one that is not. The output shown in Fig. 4d demonstrates that including an ITP search
to solve cubic polynomials removes discontinuities that occur in $Cl^-$, $NO_3^-$, $H^+$ and $NH_3$ near 1.4 mol m$^{-3}$ – and hence these
species now show a smooth transition over the entire range of TS. HETP has a limiting precision of $1\times10^{-28}$ mol m$^{-3}$, which
is the likely cause of the $HSO_4^-$ concentration becoming zero in Fig. 4(c-d) when TS is between about $2.15\times10^{-12}$ and $2.4\times10^{-12}$ mol m$^{-3}$.


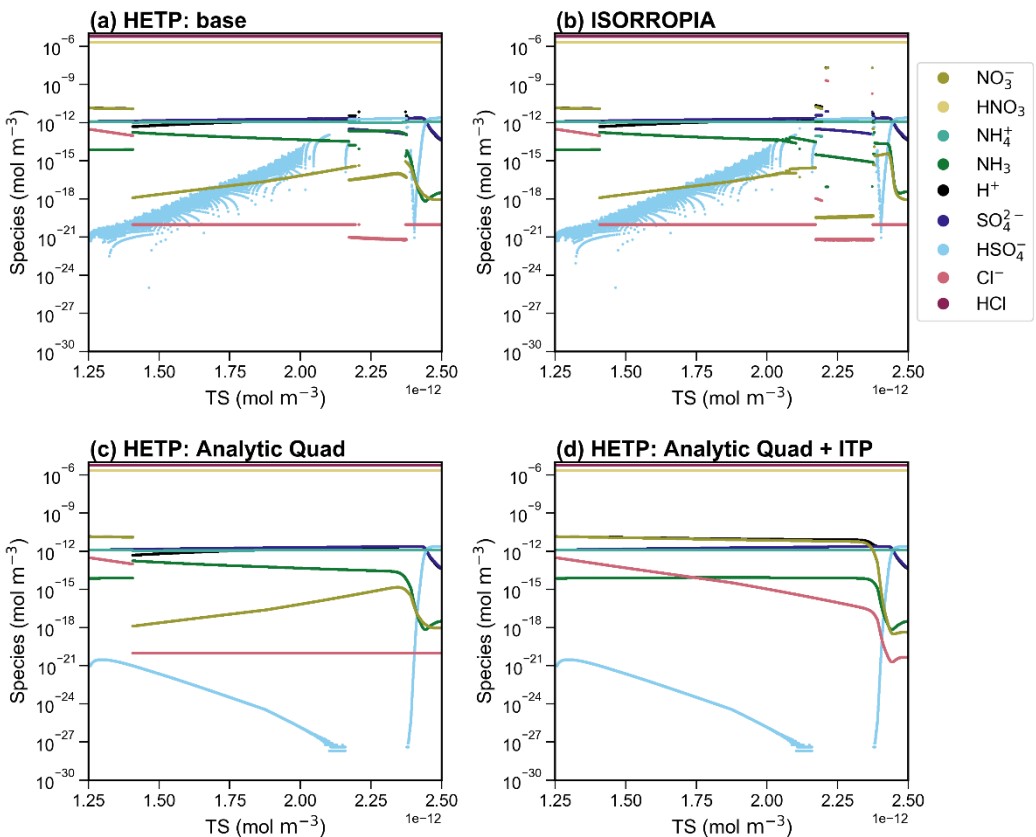

**Figure 4:** A side-by-side comparison of the output from HETP (a, c, d) and ISORROPIA (b) for CALCI6. In (a) HETP does not include
any methodological improvements to polynomial root calculations. In (c) HETP uses an updated methodology to calculate polynomial roots.
In (d), HETP uses an updated methodology to calculate polynomial roots, as well as an ITP search to determine cubic polynomial roots.
ISORROPIA shown in (b) solves quadratic equations using the 'standard' quadratic formula and attempts to find an exact analytic solution
of cubic equations. All input precursor species are held constant, except the total available sulfate (TS) which is varied over 10,000 sets of
initial conditions. The air temperature and relative humidity are 243 K and 5% respectively, for all test cases in the figure. The convergence
criteria are consistent between both solvers (see text).

## 4.2 Comparison using input from the GEM-MACH air-quality model

Aside from generating artificial sets of input data to evaluate HETP (Sect. 4.1), the value of which is to demonstrate relative solution stability across small increments in input conditions, a comparison between HETP and ISORROPIA can be completed using more realistic input conditions obtained from the GEM-MACH air-quality model (Makar et al., 2018). In this section, 20,000 unique sets of input data ('test cases') from GEM-MACH are investigated for each chemical subspace, with 10,000 test cases obtained from summer days and 10,000 test cases obtained from winter days. These test cases were selected from input conditions generated from a 10 km resolution simulation with a domain covering North America. The test cases were chosen randomly so that the selected set of test cases spans across a broad range of temperatures and relative humidity, typical of actual tropospheric conditions. Table 4 gives the relative frequency of calls to each chemical subspace as a percentage of the total calls in GEM-MACH determined from four days (2 in the winter and 2 in the summer). It should be noted that subspaces A2, B4 and C2 all require that TN be formally zero. A low number limit in the GEM-MACH model prevents true zero conditions from occurring, hence the given subroutines are not called in this practical application test. The majority of calls are to the subspaces O7, M8 and L9 which comprise more than 75 % of the total calls on these four days. Therefore, most situations encountered in GEM-MACH over North America have a non-zero amount of base cation species present ($K^+$, $Mg^{2+}$ and $Ca^{2+}$).

**Table 4**: The percentage of total calls to each subspace determined from four separate days (2 in the winter and 2 in the summer). The call frequencies are determined from the 10 km domain of the GEM-MACH air quality model which covers all of North America. Any subspace with > 10 % of total calls is bolded in the table.

| Case | A2 | B4 | C2 | D3 | E4 | F2 | G5 | H6 | I6 | J3 | O7 | M8 | P13 | L9 | K4 |
|---|---|---|---|---|---|---|---|---|---|---|---|---|---|---|---|
| % Called | 0.000 | 0.000 | 0.000 | 9.735 | 4.470 | 0.016 | 2.479 | 0.709 | 3.825 | 0.038 | **31.72** | **25.85** | 0.044 | **20.88** | 0.232 |

Figure 5 displays a scatter plot of $Cl^-$ and HCl (left panels) and $NO_3^-$ and $HNO_3$ (right panels) output from ISORROPIA (y-axis) and HETP (x-axis). Fig 5(a-b) displays CALCM8:summer (hereafter M8:S) and Fig. 5(c-d) shows CALCG5:winter (hereafter G5:W). The black dashed lines give a one-to-one relationship, denoting where HETP and ISORROPIA agree exactly. There is relatively good agreement between the two solvers for M8:S, despite the differences noted for this subspace in Sect 4.1. However, for G5:W a large amount of scatter exists, demonstrating disagreement between the two solvers for some test cases. This disagreement is likely related to the choice of root-finding method and/or other numerical updates that have been made to the HETP code, as described in Sect. 3. The differences between the two solvers noted for $Cl^-$ and HCl in Fig. 5(c) are only for very low concentrations, which likely would not be impactful in practical air-quality applications.

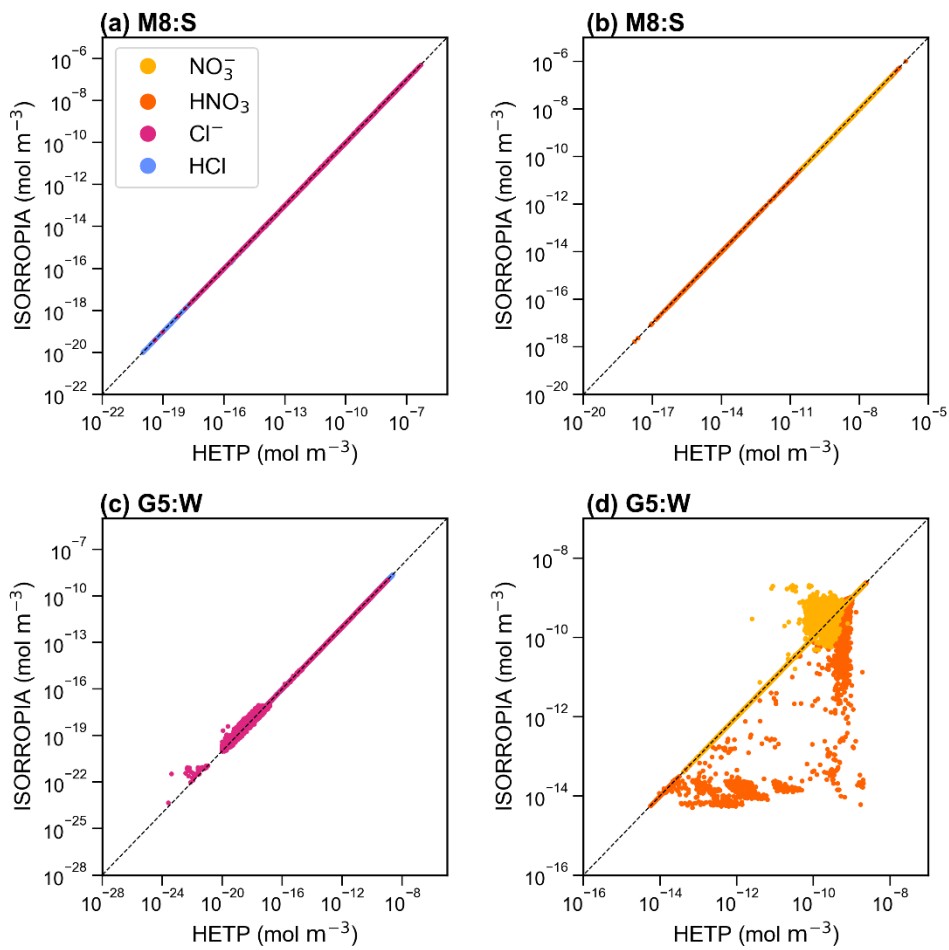

**Figure 5:** A scatter plot of the output concentrations (mol m$^{-3}$) from ISORROPIA (y-axis) compared against HETP (x-axis) for M8:Summer (M8:S) (a,b) and G5:Winter (G5:W) (c,d), calculated from 10,000 input test cases obtained from the GEM-MACH air-quality model. The solid black line gives a one-to-one relationship. Speciation is given in the legend shown in panel a.

As in Sect. 4.1, statistics of $\xi'$ are calculated from the output of each solver to judge the accuracy of the equilibrium

solution. This is especially important since the test cases in this section cannot be plotted in a regular fashion (as in Sect. 4.1),

to graphically reveal obvious numerical instabilities. Figure 6 displays a box and whisker plot of $\xi'$ for the chemical subspaces

G5, H6, O7, M8 and P13. These subspaces all require bisection or ITP and must have chloride present, with $K_{HCl}$ providing

the 'final convergence check' (except for H6). The statistics shown in Fig. 6 include the data shown in Fig. 5 for subspaces

M8:S and G5:W. Fig. 6(a) and (b) show $\xi'$ for $K_{HCl}$ and $K_{HNO_3}$ respectively. Each panel shows $\xi'$ for both seasons with

summer having a ':S' label and winter having a ':W' label. In the box plot, the 25$^{th}$ percentile, median and 75$^{th}$ percentile of

$\xi'$ correspond to the bottom of the box, center line in the box, and top of the box respectively. The bottom and top whisker of each box gives the minimum and maximum of $\xi'$ respectively; if the bottom whisker extends off the graph, then the minimum $\xi'$ is zero. Except for G5:S, H6:S, H6:W, M8:S and M8:W, the median $\xi'$ of $K_{HNO_3}$ is smaller in HETP than ISORROPIA for all subspaces shown in Fig. 6b. For $K_{HCl}$, all subspaces except H6:S and H6:W have a smaller median $\xi'$ in HETP than
ISORROPIA. We note that despite HETP having lower median $\xi'$ than ISORROPIA for some subspaces, the magnitude of $\xi'$ suggests that ISORROPIA is nevertheless providing sufficiently accurate output for most test cases. For the input data investigated here, the subspace H6 is performing poorly in both solvers, with a median $\xi' > 0.5$ for all equilibrium constants (but with marginally worse performance in HETP than in ISORROPIA). For example, in H6:S for $K_{HNO_3}$ the 75th percentile in HETP is 31.8, and in ISORROPIA it is 13.4. H6 is unique relative to the other subspaces requiring a root-finding method
(i.e., G5, O7, M8 and P13), since the objective function used to determine the root of the system of equations does not include $H^+$ explicitly. The expressions for $\xi'$ used in Fig. 6, however, explicitly evaluate the convergence of $H^+$ relative to $K_{HCl}$ and $K_{HNO_3}$ equilibria. The relatively poor performance of the H6 algorithm when evaluated using $\xi'$ thus tells us that although the other ions and gases in the H6 chemical subspace have converged with the existing solution procedure, convergence with respect to $H^+$ remains poor.

Returning to the scatter noted in $HNO_3$ and $NO_3^-$ between the two solvers in G5:W (Fig. 5d), it is clear from the statistics of $\xi'$ for $K_{HNO_3}$ and $K_{HCl}$ shown in Fig. 6 that both solvers are producing output that spans a broad range of accuracy. The 75th percentile of $K_{HNO_3}$ and $K_{HCl}$ are 2 orders of magnitude lower in HETP than ISORROPIA. For $K_{HCl}$ the 75th percentile of $\xi'$ is $6.93 \times 10^{-2}$ and 4.35 in HETP and ISORROPIA respectively. However, the maximum $\xi'$ are a similar magnitude in each solver. This suggests that both solvers are struggling with partitioning between the aqueous and gaseous phase for some
test cases investigated here. Of the 10,000 test cases analyzed in G5:W, 14.02 % are identified in HETP as having 'oscillatory behavior' (see Sect. 3, point 3). These flagged test cases generally have large $\xi'$ for all equilibrium constants (in both solvers), which is related to poor convergence during the iterative process. Removing these flagged test cases reduces the median and 75th percentile of $\xi'$ (for $K_{HNO_3}$ and $K_{HCl}$) by an order of magnitude in both solvers. For HETP the median $\xi'$ for $K_{HNO_3}$ reduces to $4.89 \times 10^{-8}$ (from $4.60 \times 10^{-7}$) and for ISORROPIA the median $\xi'$ reduces to $2.72 \times 10^{-6}$ (from $5.59 \times 10^{-5}$). The modification
to account for 'oscillatory behavior' has the effect of reducing $\xi'$ for the flagged test cases in HETP compared to ISORROPIA – for the 14.02 % of test cases affected, the median $\xi'$ for $K_{HNO_3}$ is 0.28 for HETP, but for ISORROPIA it is 2.65. Furthermore, 75.3 % of the flagged test cases are times when $Cl^-$ is predicted to be $< 1 \times 10^{-16}$ mol m$^{-3}$ ($Cl^-$ is the bisected variable in G5), and all flagged test cases have TCl $< 1 \times 10^{-10}$ mol m$^{-3}$. For test cases where the output from each solver agrees well and falls along the one-to-one line in Fig. 5c-d, $\xi'$ for $K_{HNO_3}$ and $K_{HCl}$ are minimized in each solver. The statistics of $\xi'$ for other
subspaces not discussed here are summarized in Table S4 (summer) and Table S5 (winter) of the supplemental information.

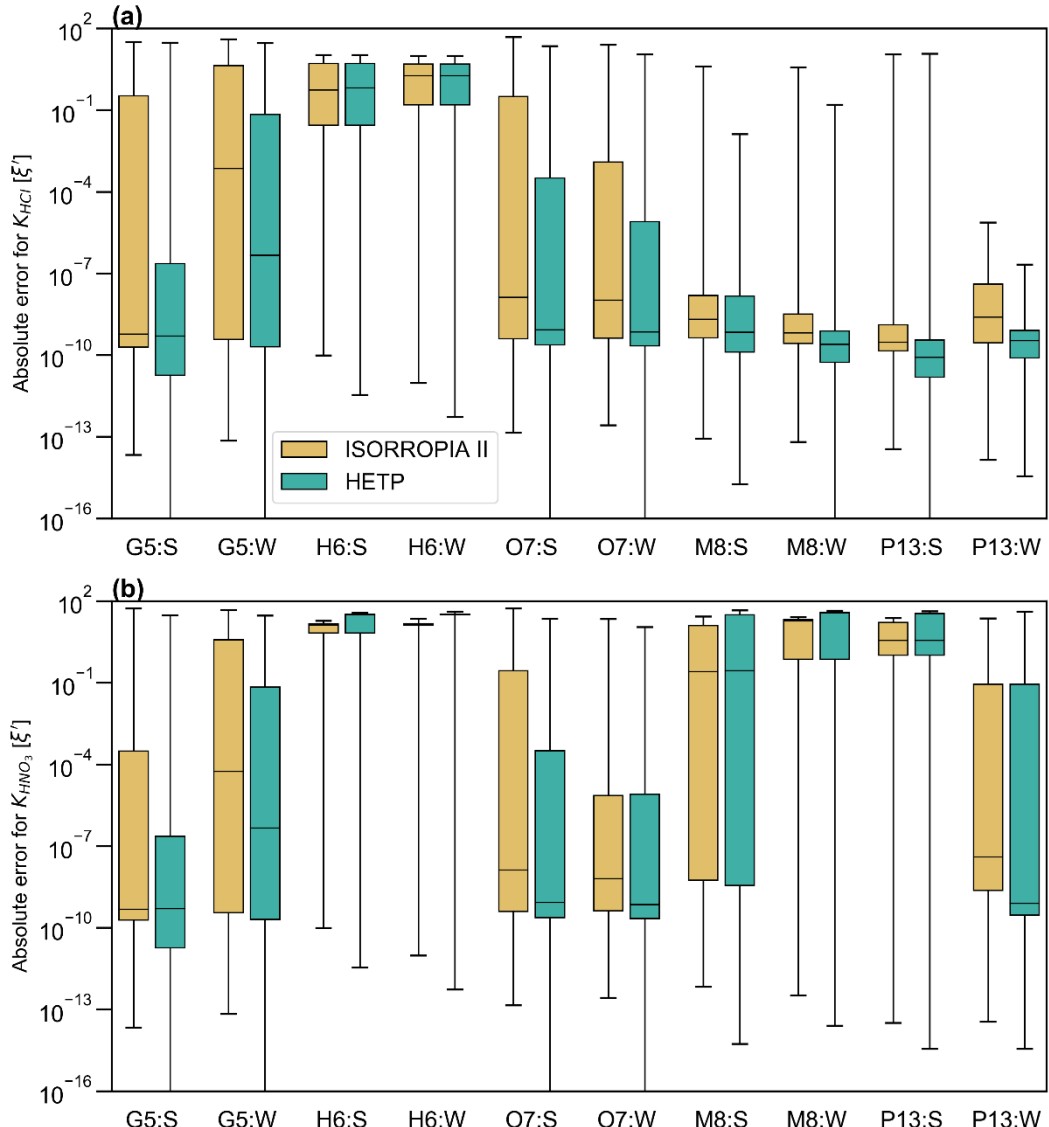

**Figure 6:** A box and whisker plot of the absolute error $\xi' = |\log(K_{calc}) - \log(K_{true})|$ for (a) $K_{HCl}$ and (b) $K_{HNO_3}$. The summer season is denoted by ':S' and the winter season is denoted by ':W' in the x-axis labels. $\xi'$ is calculated from a set of 10,000 test cases in each season (obtained from the GEM-MACH air-quality model). $\xi'$ shown in the figure for M8 and G5 correspond to the scatter plots shown in Fig. 5. The median $\xi'$ is represented by the solid black line in the center of each box, and the 25th and 75th percentiles correspond to the bottom and top of each box respectively. The whiskers give the maximum (top) and minimum (bottom) of $\xi'$.


### 4.3 Computational time

The mean time (determined from 10 repeated samples) required for the central processing unit (CPU) of a Lenovo SV650v2 DWC computer to solve the test cases from Sect. 4.2 for each season and subspace are given in Table 5. The timing tests have an estimated uncertainty of $\pm 1$ %. For HETP, two sets of timing tests are reported. Test 1, labelled '$T_{HETV}$', refers to timing using a global convergence criterion for all tests within a given chemical subspace, representing a "vectorized" test where all $n$ test cases for a given subspace are solved simultaneously. This is the methodology used in Makar et al., (2003), where the

great reduction in processing time associated with vectorization on a vector compiler was used to offset the fact that the number of iterations was determined by the single test case with the worst convergence behavior. Test 2, labelled '$T_{HETP}$', refers to a case-by-case test where the solver is called individually for each test case (i.e., the solver is called $n$ times). In the latter test, the time associated with subroutine calls is offset by the number of iterations becoming test specific. The first strategy may be more efficient, aside from vectorization architecture gains, when the convergence criteria are relatively similar across grid-

cells, that is, all input problems converge with the same number of iterations. The second strategy may be more efficient when the distribution of convergence is more heterogeneous, with some test cases requiring many more iterations than others. ISORROPIA ($T_{ISO}$) requires a case-by-case implementation and cannot solve $n$ cases simultaneously. The convergence criteria are identical to those used in the previous sections (Sect. 4.1 and 4.2). In the case of ISORROPIA, it is important to reaffirm that the '-r8' flag was used during compilation, forcing all calculations to be performed in double precision (as in the default

implementation of HETP) and removing precision as a possible cause for differences in performance. For the subspaces D3, G5, H6, O7, M8 and P13 all test cases investigated were chosen so that they require the application of a root-finding method for convergence, since these are the most computationally intensive cases encountered by the solver. As noted above, not all chemical subspaces have 10,000 unique input data derived from GEM-MACH simulations for the days sampled from each season. Specifically, in the winter the subspaces A2, B4, C2 and F2 do not have enough suitable input data from which to

draw 10,000 unique samples, and likewise in the summer for subspaces A2, B4 and C2. For winter, input data from J3 are used for F2, except with TNa = 0 and TCl = 0, the aim here being to provide timing tests across a realistic range of initial conditions. It should be noted again that the subspaces A2, B4 and C2 were not executed by GEM-MACH on either day for the reasons noted in Sect. 4.2. Therefore, like F2 (winter), the input data used to analyze D3, E4 and F2 are used to analyze A2, B4 and C2 respectively, except with TN = 0.



**Table 5:** The average computational time ($T$) (calculated from 10 samples) required to solve 10,000 unique sets of input conditions (from summer and winter), using ISORROPIA ($T_{ISO}$), the vectorized solver of HETP ($T_{HETV}$) and the case-by-case solver of HETP ($T_{HETP}$). Input conditions were obtained from the GEM-MACH air-quality model, and the convergence criteria are consistent between both solvers (see text). The speed up is a dimensionless quantity, with the non-bracketed value representing $T_{ISO}/T_{HETV}$ and the bracketed value representing $T_{ISO}/T_{HETP}$. A value > 1 implies that HETP (or HETV) is computationally faster, while a value < 1 implies that ISORROPIA is computationally faster. In the first three columns of each season, the bolded value denotes the fastest execution time between each of the solvers. The bolded value in the speed up column shows which solver style is computationally faster (i.e., HETP or HETV); an underlined value in this column signifies that HETV is computationally slower than ISORROPIA for that subcase (row).

| Subroutine | Winter | | | | Summer | | | |
|---|---|---|---|---|---|---|---|---|
| | $T_{HETV}$ (s) | $T_{HETP}$ (s) | $T_{ISO}$ (s) | Speed up | $T_{HETV}$ (s) | $T_{HETP}$ (s) | $T_{ISO}$ (s) | Speed up |
| CALCA2 | 0.044 | **0.042** | 0.061 | 1.39 (**1.45**) | 0.049 | **0.046** | 0.069 | 1.41 (**1.50**) |
| CALCB4 | **0.011** | **0.011** | 0.022 | **2.00** (**2.00**) | **0.011** | **0.011** | 0.022 | **2.00** (**2.00**) |
| CALCC2 | 0.010 | **0.009** | 0.020 | 2.00 (**2.22**) | **0.010** | **0.010** | 0.020 | **2.00** (**2.00**) |
| CALCD3 | 0.335 | **0.270** | 0.486 | 145 (**1.80**) | 0.347 | **0.262** | 0.461 | 1.33 (**1.76**) |
| CALCE4 | **0.013** | **0.013** | 0.027 | 2.08 (**2.08**) | **0.014** | **0.014** | 0.026 | **1.86** (**1.86**) |
| CALCF2 | 0.013 | **0.012** | 0.024 | 1.85 (**2.00**) | 0.013 | **0.012** | 0.024 | 1.85 (**2.00**) |
| CALCG5 | 0.447 | **0.373** | 0.806 | 1.80 (**2.16**) | 0.360 | **0.284** | 0.704 | 1.96 (**2.48**) |
| CALCH6 | 0.126 | **0.059** | 0.108 | 0.86 (**1.83**) | 0.136 | **0.068** | 0.121 | 0.89 (**1.78**) |
| CALCI6 | 0.027 | **0.026** | 0.037 | 1.37 (**1.42**) | 0.029 | **0.027** | 0.039 | 1.34 (**1.44**) |
| CALCJ3 | **0.030** | **0.030** | 0.039 | **1.30** (**1.30**) | 0.032 | **0.031** | 0.041 | 1.28 (**1.32**) |
| CALCO7 | 0.690 | **0.541** | 1.202 | 1.74 (**2.22**) | 0.688 | **0.558** | 1.262 | 1.83 (**2.26**) |
| CALCM8 | 0.409 | **0.259** | 0.607 | 1.48 (**2.34**) | 0.482 | **0.354** | 0.646 | 1.34 (**1.82**) |
| CALCP13 | 0.409 | **0.290** | 0.727 | 1.78 (**2.51**) | 0.376 | **0.184** | 0.796 | 2.12 (**4.33**) |
| CALCL9 | 0.041 | **0.039** | 0.059 | 1.44 (**1.51**) | 0.038 | **0.036** | 0.058 | 1.53 (**1.61**) |
| CALCK4 | 0.044 | **0.042** | 0.063 | 1.43 (**1.50**) | 0.042 | **0.040** | 0.060 | 1.43 (**1.50**) |
| Sum of GEM–MACH tests | 2.65 | **2.02** | 4.29 | 1.62 (**2.12**) | 2.63 | **1.94** | 4.35 | 1.65 (**2.24**) |

The CPU timing results demonstrate that all subspaces (except H6, winter and summer) execute faster in HETP's vectorized $T_{HETV}$ implementation than ISORROPIA. In some cases the speed up is significant: for CALCO7 the speed up is about a factor of 1.74 to 1.83 when using $T_{HETV}$. An even greater speed up can be achieved by using the case-by-case $T_{HETP}$

implementation for some subspaces, specifically those that require bisection (A2, D3, G5, H6, O7, M8 and P13). Unlike $T_{HETV}$, *all* chemical subspaces execute faster in $T_{HETP}$ than ISORROPIA. For the sets of test cases investigated in this work, the best-case performance is found in P13:S, where $T_{HETV}$ executes in about 0.38 s, but $T_{HETP}$ executes in about 0.18 s (the latter being about ~4.3x faster than ISORROPIA). The speed up afforded by HETP for this subcase is largely the result of HETP's root-

finding methodology (ITP), which requires fewer iterations on average to obtain a solution with an equivalent or better level of accuracy as ISORROPIA. The statistics related to the number of iterations required by the root-finding methodology of each solver to achieve convergence of the major systems are given in Table 6, for the same input data used to generate the timing tests shown in Table 5. For P13:S which has the best-case performance, ITP in HETP requires on average 8.0 iterations for convergence, while bisection in ISORROPIA requires on average 42.5 iterations. Thus, HETP's root-finding method

requires about 19 % of the iterations required by ISORROPIA for this set of input conditions, while executing in about 23 % of the time using the case-by-case mode. The overall performance for the tests in GEM-MACH (bottom row of Table 5) show the average performance of HETP operating in the case-by-case mode results in a speed up relative to ISORROPIA of a factor of 2.24x for the summer tests, and 2.12x for the winter tests. The inclusion of an ITP search for the smallest positive real root of cubic equations in I6, J3, L9 and K4 substantially increases the execution time of the solver for these chemical subspaces

relative to no ITP search, but despite this, HETP still executes in less time than ISORROPIA for these subcases.

The difference between $T_{HETP}$ and $T_{HETV}$ becomes even more apparent, and in favor of $T_{HETP}$, if a significant amount of test cases do not require bisection. While $T_{HETV}$ includes a return statement to reorder the problem, removing those test cases that have converged or have no solution prior to entering ITP, the root bracketing stage in $T_{HETV}$ will nonetheless need to be repeated a second time for all test cases that do require ITP. Note that the root bracketing stage identifies an interval

where the objective function has a sign change. Assuming a continuous function, this sign change signifies that a root exists within the interval. Furthermore, in $T_{HETV}$ some test cases may iterate in the root-bracketing stage more times than necessary (i.e., one test case has an identified interval, but other test cases within the same chemical subspace being solved by a global convergence criterion do not), thereby introducing excess computations into $T_{HETV}$ that do not exist in $T_{HETP}$. This is especially true as the variable $ndiv$, which controls the number of subdivisions searched for a sign change, is increased. Thus, in most

applications, and for the computer architecture tested here, the case-by-case $T_{HETP}$ implementation will be preferred. Both options are available as separate versions of code, and we recommend users test both options of the code on their own system to determine the best performance.

The results presented herein have demonstrated that HETP is able to provide output for these subspaces that is more accurate overall, while executing up to 4.3x faster than ISORROPIA, with an average performance increase in a practical

application between 2.12x and 2.24x (using the case-by-case mode). The subspace H6 which executes slower in $T_{HETV}$ than $T_{HETP}$, and is also less accurate than ISORROPIA for most input test cases, accounts for < 1 % of the test cases on the days sampled (see Table 4).

**Table 6:** Statistics describing the number of iterations required to achieve convergence of bisection (ISORROPIA) or ITP (HETP) for the timing tests shown in Table 5. The final column shows the average speed up, calculated for each row as the mean number of iterations from ISORROPIA divided by the mean number of iterations from HETP.

| | ISORROPIA: Winter | | | | | | HETP: Winter | | | | | | |
|---|---|---|---|---|---|---|---|---|---|---|---|---|---|
| **Case** | **Median** | **Q25** | **Q75** | **Min** | **Max** | **Mean** | **Median** | **Q25** | **Q75** | **Min** | **Max** | **Mean** | **Speed up** |
| CALCD3 | 29 | 28 | 43 | 28 | 61 | 33.8 | 29 | 10 | 41 | 5 | 61 | 26.7 | 1.27 |
| CALCG5 | 34 | 32 | 35 | 28 | 52 | 34.3 | 30 | 7 | 34 | 1 | 51 | 22.7 | 1.51 |
| CALCH6 | 33 | 30 | 37 | 28 | 41 | 33.6 | 7 | 7 | 7 | 4 | 37 | 9.9 | 3.39 |
| CALCO7 | 34 | 32 | 37 | 27 | 52 | 35.2 | 14 | 8 | 34 | 1 | 53 | 20.6 | 1.71 |
| CALCM8 | 31 | 30 | 33 | 28 | 43 | 31.6 | 7 | 7 | 11 | 4 | 38 | 11.3 | 2.80 |
| CALCP13 | 31 | 28 | 35 | 28 | 58 | 31.8 | 8 | 7 | 9 | 5 | 42 | 11.9 | 2.67 |
| | ISORROPIA: Summer | | | | | | HETP: Summer | | | | | | |
| **Case** | **Median** | **Q25** | **Q75** | **Min** | **Max** | **Mean** | **Median** | **Q25** | **Q75** | **Min** | **Max** | **Mean** | **Speed up** |
| CALCD3 | 29 | 28 | 32 | 22 | 68 | 33.5 | 28 | 24 | 31 | 1 | 62 | 28.2 | 1.19 |
| CALCG5 | 33 | 31 | 36 | 26 | 57 | 34.1 | 12 | 7 | 32 | 1 | 44 | 17.6 | 1.94 |
| CALCH6 | 32 | 28 | 37 | 28 | 40 | 32.6 | 7 | 7 | 29 | 5 | 38 | 15.1 | 2.16 |
| CALCO7 | 34 | 32 | 36 | 28 | 47 | 33.9 | 17 | 9 | 34 | 5 | 45 | 21.1 | 1.61 |
| CALCM8 | 32 | 29 | 34 | 28 | 41 | 31.9 | 11 | 7 | 29 | 5 | 40 | 18.2 | 1.75 |
| CALCP13 | 42 | 39 | 45 | 28 | 60 | 42.5 | 7 | 6 | 7 | 5 | 61 | 8.0 | 5.31 |

## 5 Conclusions

In this work we have presented HETP, an updated solver to perform thermodynamic equilibrium calculations of the $H^+$–$SO_4^{2-}$–$NH_4^+$–$NO_3^-$–$Cl^-$–$Na^+$–$Ca^{2+}$–$K^+$–$Mg^{2+}$–$H_2O$ chemical system, based on the algorithms of ISORROPIA in the forward metastable state. HETP has been updated in several ways to improve both the computational speed and accuracy compared to ISORROPIA. For most input conditions HETP produces equivalent results to ISORROPIA, but for some input conditions the output from the solvers can diverge. Analysis of the output from each solver suggests that HETP's use of ITP, instead of bisection, improves the accuracy of its equilibrium solution for some input conditions by obtaining a more accurate initial estimate of the root prior to the commencement of the ITP search. At the same time, ITP can reduce the number of iterations required for convergence. The differences may be formally linked to reduced accuracy of the ISORROPIA solver's output due to several numerical issues as described in the sections above. In addition to providing more accurate output for most test cases, HETP, when implemented to solve $n$ test cases simultaneously, may execute 1.3 to 2.1 times faster than ISORROPIA (except for CALCH6), based on input from the CTM GEM-MACH. Alternatively, when HETP is implemented as a case-by-

case solver (the solver is called $n$ times), then HETP is 1.3 to 4.3 times faster than ISORROPIA for individual chemical subspaces, and 2.1 to 2.2 times faster than ISORROPIA on average, with the speed-up being most significant in subspaces that require the application of a root-finding method for convergence.

## 6 Code and data availability

The data used in the analysis presented herein, and the HETP code, are available online at https://doi.org/10.5281/zenodo.8164704 (Miller, 2024).

## 7 Author contribution

SJM developed the HETP code, performed the analysis presented herein, and wrote the manuscript, with significant contributions from PAM. PAM also assisted with the development of the HETP code. CJL assisted with inorganic heterogeneous chemistry module interfacing with GEM-MACH, GEM-MACH tests, and incorporation of the new module into the regional transport model.

## 8 Competing interests

The contact author has declared that none of the authors has any competing interests.

## 9 Acknowledgements

This work was funded under the Oil Sands Monitoring (OSM) Program, subproject "Integrated Atmospheric Deposition", subproject A-PD-6-2324. It is independent of any position of the OSM Program. We would like to thank Wanmin Gong for reviewing the manuscript prior to submission for peer-review and Verica Savic-Jovcic for providing advice during the peer-review process.

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
