# Peer review of "HETerogeneous vectorized or Parallel (HETPv1.0): An updated inorganic heterogeneous chemistry solver for metastable state NH4+– Na+–Ca2+–K+–Mg2+–SO42––NO3––Cl– based on ISORROPIA II"

_Geoscientific Model Development, 2023_

## Referee Comment (RC3)

**Review of "HETerogeneous vectorized or Parallel (HETPv1.0): An updated inorganic heterogeneous chemistry solver for metastable state NH$_4^+$–Na$^+$–Ca$^{2+}$–K$^+$–Mg$^{2+}$–SO$_4^{2-}$–NO$_3^-$–Cl$^-$ based on ISORROPIA II " for *Geoscientific Model Development***

**General Comments**

Miller, Makar and Lee describe the development of a novel computational model written in Fortran 90 for the thermodynamic partitioning of a total amount of inorganic species as listed in the title between the gas and liquid phases in the presence of water. Typically, they follow the algorithms of the forward solution of ISORROPIA II for the metastable cases. Additionally, they identify and correct algorithmic formulations that lead to errors in the output of ISORROPIA. Furthermore, they implement a recently developed root finding approach instead of bisection that improves the accuracy of equilibrium solutions and the speed of arriving at them in most cases. The model is thoroughly tested not only against ISORROPIA but against the analytical solution of the equilibrium equations. Additionally, the accuracy and computational costs were explored for realistic cases as derived from the regional GEM-MACH model for selected conditions summer and winter. Finally, the code for this open source model has been made publicly available through Zenodo.

The manuscript is replete with evidence of the carefulness with which this model was developed and tested. The authors are clear as to where HETP outperforms ISORROPIA in terms of accuracy or computational cost but also where it underperforms, which is less common. In most cases, the authors posit fundamental explanations for why HETP performs differently than ISORROPIA, identifying specific algorithmic changes associated with the results in most tables and figures. For instance, the third subtable in Table 2 is evidence of an excellent investigation of the algorithmic explanation for the observed improvements. The authors have helpfully structured the manuscript by selecting tests that pertain to specific algorithmic changes to avoid inundating the reader with excessive comparisons while still demonstrating the extent of the testing conducted and then building to the comprehensive timing tests, which are sufficient to demonstrate the robustness of the model development process.

The authors are helpfully straightforward about HETP solving only the metastable cases. Since most chemical transport models only use the metastable solutions from ISORROPIA, this clarity is important but not impactful for those inclined to adopt the model. The primary change I think would improve the manuscript is to be clear at least in the abstract if not in the title that only the forward solution in ISORROPIA is included. Currently, the first mention that HETP only addresses the forward solution of ISORROPIA is at the beginning of Section 2 though in the introduction ISORROPIA is referenced as being used in CMAQ, in which both the forward and reverse solutions are employed for dynamic equilibrium of coarse mode aerosol with the rest of the population. So as to not cause a reader undue hope that well-documented issues with the reverse solution have been resolved with HETP, I would urge the authors to consider being clear that only the forward solution is implemented in HETP.

With this small but important change, a few responses to specific comments aimed to add value for future readers, and a careful grammatical revision including reduction of the number of parenthetical phrases, I would expect that the publication of this

manuscript would benefit many in the atmospheric modeling community for years to come.

**Specific Comments**

Lines 9–12    The claims in the first two lines of the introduction are very important and are well-supported in the manuscript. The current language in the first two sentences, especially the second, obscures the value of this paper somewhat. Consider rephrasing these two sentences, possibly into three shorter ones, for the sake of clarity and impact.

Lines 80-92    The authors have reasonable explanations for the use of the metastable assumption documented. One important additional reason is that the history of the aerosols is not tracked in these models such that one cannot know whether the mutual efflorescence or deliquescence relative humidity would best characterize the conditions for crystallization.

    Also, consider starting a new paragraph with the metastable state discussion since this topic is slightly distinct from the thesis of the paragraph and other content in it.

Lines 116-8    Consider making a separate sentence with the content beginning with "which" on account of the parenthetical nature of the descriptions of parallel and vector implementations not being sufficiently clear. If that restructuring is not desired, at least make the parenthetical explanations parallel in form and each properly introduced.

Lines 305-18    Many of the enumerated algorithmic improvements, such as the more robust solution of cubic equations, are documented with sufficient detail. One helpful addition to these enumerated algorithmic improvements would be to identify one example of their application by line number in the HETP.ftn90 file included in the Zenodo repository.

**Grammatical Comments**

Line 45    Elsewhere "N" is written as an italicized variable. Consider formalizing this expression, too.

Line 115    Please ensure that the "forward" nature of the solution is somehow mentioned in this helpful, governing sentence.

Line 119    It is not clear to me where the idea of the metastable state representing the "efflorescence branch" arose. To my knowledge, the mutual deliquescence relative humidities are used in ISORROPIA and insufficient information exists for the efflorescence relative humidities of these salt mixtures to treat efflorescence in a thermodynamic equilibrium model of inorganic aerosols. It would be sufficient to delete this parenthetical phrase and leave the reader to the other portions of the manuscript in which the metastable state is more accurately described.

| | |
|---|---|
| Line 159 | Here and in some other cases, a ";" is missing: "initial mass adjustments, however any output" should be "initial mass adjustments; however, any output". Please revise here and in other places as needed. |
| Line 193-4 | No comma exists after the adverb at the beginning of the sentence (i.e., "Currently,") but an unnecessary comma is used before "and" in the same sentence though the conjunction is not followed by an independent clause. Elsewhere, commas are used where semicolons are needed. Please revise the use and absence of commas throughout the manuscript. |
| Line 195-9 | Four independent clauses are joined by two semicolons and one conjunction that is not preceded by a comma. Please look for run-on sentences such as these and revise as needed. |
| Line 230 | "the ZSR correlation" would be better as "the Zdanovskii-Stokes-Robinson (ZSR) correlation". |
| Line 271 | Although the code-based expression "TCl = max(TCl, $1\times10^{-10}$)" will be comprehensible to those accustomed to writing Fortran, perhaps using words to express the concepts would be more appropriate for this part of the text. |
| Line 706 | "computations into in" was likely meant to be "computations into". |

---

## Author Comment (AC1)

**This document is structured with each referee or community comment (in orange) followed by an author response and changes to the manuscript (in blue). The specific changes made to the text of the manuscript are bolded in the authors' responses for emphasis.**

**Line numbers in the authors' responses refer to the revised manuscript with "tracked changes" turned off.**

**CC#1**

This is an exciting development and addresses an issue with ISORROPIA II which has been previously noted in the GEOS-Chem community (https://wiki.seas.harvard.edu/geos-chem/index.php/ISORROPIA_II#Investigating_persistent_noise_observed_in_ISORROPIA_output). We have found implementation of HETP into GEOS-Chem as an alternative to ISORROPIA II to be straightforward (https://github.com/sdeastham/geos-chem/tree/feature/HETP) and to produce results which are very close to those from ISORROPIA II, with no appreciable speed penalty. This has also resulted in a reduction in noise when evaluating the effects of (e.g.) aviation emissions on surface $PM_{2.5}$.

- We would like to thank Sebastian Eastham for taking the time to incorporate HETP into GEOS-Chem, as well as providing this valuable and positive feedback to us.

That having been said, HETP would benefit from two additional changes in terms of code:

1. The code currently includes a large number of evaluations such as `variable == .true.`. This is not supported in standard Fortran (see e.g. section 10.1.5.5.1 in the Fortran 2023 standard final draft, publicly accessible at https://j3-fortran.org/doc/year/23/23-007r1.pdf, which explicitly states that this operation is invalid) and gfortran will not compile it. However, the simple fix - replacing the incorrect == or `.eq.` with `.eqv.` - does not work because `.eqv.` has a lower precedence (see e.g. https://stevelionel.com/drfortran/2000/04/29/doctor-fortran-in-to-eqv-or-to-neqv-that-is-the-question-or-its-only-logical/), resulting in incorrect logical evaluations. The code needs to be updated to comply with the Fortran standards, ideally by replacing all statements of `x == .true.` and `y == .false.` with simply `x` and `.not. y` respectively.
   - All logical variable statement evaluations have been updated to meet the Fortran standard throughout the code.

2. It would significantly improve the portability of the code for HETP to be package in such a way that it can be compiled as a library, or at least for the code to be entirely contained within a module (so that users can specify `use HETP, only : calculate_thermodynamics` or similar from their own code). That would allow multiple models to use the code without modification, and therefore support ongoing community development which could benefit all models equally.
   - The code for all HETP subroutines has been placed within a module file (hetp_mod.ftn90), with all subroutines declared as private, except mach_hetp_15cases.
   - Another module file (mach_hetp_mod.ftn90) holds the water activity arrays, kind declarations and convergence variables.

**Referee #1**

Stefan Miller, Paul Makar and Colin Lee describe a new and improved solver method termed HETP for solving the thermodynamic inorganic aerosol gas–liquid partitioning problem based on the ISORROPIA II model and its approach. This comprehensive study provides an overview of inorganic aerosol thermodynamics and the typically considered set of species, a description of how the ISORROPIA algorithm and its division of the thermodynamic equilibrium problem into subspaces is designed, and where, how, and why HETP differs in some of the details. It implies that the authors have acquired a detailed understanding of the ISORROPIA algorithm and potential numerical limitations, which they address to a large extent with their HETP model.

Overall, the paper is well written, the tables and figures are clear, useful, and well described, and the evaluation of HETP (and HETV) against ISORROPIA II is done in a meaningful way in terms of numerical accuracy, reliability, and computing time. This reviewer appreciates the distinct sets of comparisons discussed in Sections 4.1 and 4.2, with the latter drawing realistic examples of species inputs from an air quality model (GEM-MACH) run for the North America domain. I find this work to be a valuable effort both in terms of this article and associated evaluations and documentation, as well as in terms of the accompanying Fortran source code, which should enable relatively easy adoption into various air quality models.

My comments are minor. They should help the authors to further clarify a few details of this work prior to final publication.
- We would like to thank Referee #1 for their detailed and careful review of our work, as well as the positive feedback provided above. We feel Referee #1's comments have resulted in a much-improved manuscript.

**General comments:**

Abstract and Section 3, a wording detail: The HETP code is described as being written in "FORTRAN 90". This is a bit of an oddity, not because it is coded in Fortran, which is perfectly adequate for this computer program, but because of the spelling and implied unnecessarily narrow meaning of the language. If anything, the correct spelling is "Fortran 90" (not capitalized, unlike FORTRAN 77 and prior). However, why use/refer to the code as being in Fortran 90 and why not 95, 2003, 2008 or a more recent version of the standard of the language, such as Fortran 2018 or the upcoming Fortran 2023 standard? By that I do not mean that the authors should make use of all the latest language features, but that by writing "Fortran 90" they may simply, yet incorrectly, mean what is often referred to as "modern Fortran". That is, a Fortran version that uses free-format source code, a modular procedural (or object-oriented) programming paradigm or a combination of both, portable variable kinds, dynamic allocatable arrays, etc (see, e.g. https://fortran-lang.org). Note that the ".f90" file extension (or .ftn90 in the linked HETP package) does not specifically refer to nor restrict code to Fortran 90; rather, it is a file extension that is (to date) used for any modern free-form Fortran code (since Fortran 90). Also note that most (nearly all) of Fortran 90 is a subset of the most recent standard (since Fortran is backwards-compatible, except for a few very rare, depreciated features).

Therefore, I strongly suggest revising the sentences in this paper when Fortran 90 is mentioned, since it points to an outdated standard of the language.
- Thank you for pointing this out. We have modified the text and replaced "Fortran 90" with "modern Fortran" or just "Fortran" in the abstract and manuscript.

**Specific comments:**

1.  Line 29: "The inorganic portion …." Mentioning water as a key inorganic aerosol component would be adequate.
    - We included an explicit mention of water in this sentence, but also kept the list of inorganic species. We list the inorganic species here to familiarize a reader unfamiliar with their chemical formulae, which are used throughout the manuscript.
    - The sentence on lines 30-34 of the revised manuscript has been rewritten to: "The inorganic portion of atmospheric particulate matter consists primarily of sulfate ($SO_4^{2-}$), nitrate ($NO_3^-$), ammonium ($NH_4^+$), chloride ($Cl^-$), calcium ($Ca^{2+}$), potassium ($K^+$), magnesium ($Mg^{2+}$),  sodium ($Na^+$) **and water ($H_2O$)** (Harrison and Pio, 1983; Wang et al., 2003)."

2.  Line 32: Aside from bromide, some marine regions also show notable iodine species amounts (such as I- and IO3- ions in aerosols), e.g. Saiz-Lopez et al. (2011, Chem. Reviews).
    - Thank you for pointing this out. We now mention iodide in the text of the revised manuscript.
    - The sentence on lines 34-35 has been rewritten: "Along coastlines and within marine air masses, inorganic bromide ($Br^-$) (Sander et al., 2003) **and iodide ($I^-$) (Saiz-Lopez et al., 2011)** may also be common."

3. Line 33: "in the coarse mode"; clarify for non-experts: coarse mode of what?
   - We now specifically refer to 'coarse mode aerosols' instead of just 'coarse mode'; likewise for 'fine mode aerosols'. Furthermore, the coarse mode and fine mode have been defined in the manuscript.
   - Lines 34-35 has been modified to: "$Ca^{2+}$, $K^+$, $Mg^{2+}$, $Na^+$ and $Cl^-$ exist principally **in coarse mode aerosols (particle diameter > 2.5 μm)**, and…"
   - Lines 37-38 has been modified to: "It should be noted that a considerable amount of $K^+$ may also be present **in fine mode aerosols (particle diameter < 2.5 μm)** when it is..."

4. Line 50: Another, more recent overview of inorganic aerosol equilibrium models, including E-AIM, MOSAIC, AIOMFAC-GLE, ISORROPIA II and EQUISOLV II, is provided in the review paper by Pye et al. (2020), https://acp.copernicus.org/articles/20/4809/2020/. Perhaps worth mentioning here.
   - Thank you for pointing this out. We have now included this reference in the revised manuscript.
   - Lines 51-53 have been modified to: "Several solvers have been developed to simulate the thermodynamic partitioning of inorganic species (see Zhang et al., 2000 **and Pye et al., 2020** for a detailed review of these solvers)."

5. Line 51: Regarding rigorous solvers: another rigorous and sophisticated solver approach is the UHAERO model/method by Amundson et al. (2006), atmos-chem-phys.net/6/975/2006/, suggested to be cited here too.
   - Thank you for noting this missing reference, we have now included it in the revised manuscript.
   - Lines 53-54 have been modified to: "AIM2 (Clegg and Pitzer, 1992; Wexler and Clegg, 2002), GFEMN (Ansari and Pandis, 1999a, b) and **UHAERO (Amundson et al., 2006)** are considered the most rigorous solvers…"

6. Line 76: "the metastable state (those subsystems in which liquid water is present)". The text in parenthesis describes a side effect of aqueous inorganic aerosols in metastable state, not the state itself. It is a potentially misleading description of what metastable state means and should be improved. The metastability refers to the fact that certain cation-anion combinations (and potentially hydrates) are present in a supersaturated aqueous solution relative to a more stable state involving presence of some crystalline salts aside from a remaining aqueous solution that is saturated with respect to those salts. In metastable state mode, the thermodynamic solution approach ignores potential formation of crystalline states (perhaps with some exceptions like CaSO4).
   - We have updated to manuscript (as shown below) to reflect the excellent definition provided by the referee above.
   - Lines 79-81 have been updated to: "ISORROPIA-lite addresses the metastable state **(i.e., it assumes a supersaturated aqueous solution where crystalline states are ignored, except $CaSO_4$)**, as well as effects of organic aerosols on the partitioning of the inorganic system. "
   - Lines 86-88 have been updated to: "In the absence of these additional sources of aerosol water, the "pure" (i.e. only) inorganic aerosol thermodynamics can result in partitioning to the **'stable'** aerosol phase as only crystalline  **salts** (no ions) **or a mixture of crystalline salts and aqueous ions that are saturated with respect to the crystalline salts**."

7. Line 89: "assume a metastable state as the most likely"; yes often, but note that an aqueous inorganic phase could also be absolutely stable, not just metastable, e.g. at high RH > 80%. Except that there may be minerals present or forming in aerosols that remain crystalline (e.g. gypsum) due to extremely low solubility in water. This was also assumed in this work for CaSO4.
   - We have modified the sentence on lines 97-99: "… applications of inorganic aerosol thermodynamics within CTMs thus tend to assume a metastable state as the most likely conditions in the troposphere, **although absolutely stable aqueous aerosols are possible above the deliquescence RH**."

8. Line 96: Perhaps of interest: in ANISORROPIA and related changes to ISORROPIA Capps et al. (2012), doi:10.5194/acp-12-527-2012, have introduced a combination of bisection and Newton's method (primarily needed for the adjoint model, not for better speed).
   - Thank you for pointing out this missing reference, which is now included in the revised manuscript.
   - Lines 105-108: "**Newton's method is also used in ANISORROPIA where it is combined with the bisection method for chemical subspaces describing a neutral aerosol. ANISORROPIA performs a sensitivity analysis on each inorganic species considered in ISORROPIA (excluding $Ca^{2+}$, $Mg^{2+}$, $K^+$) with respect to the total input precursor species concentration (Capps et al., 2012 ).**"

9. Line 117: "by grid point solutions…"; It should be explained here what is meant by grid point in this context. Is this about a grid point in a 3D air quality model?
   - We were referring to two options of (1) gathering all similar cases together and solving with a global convergence criterion versus (2) solving for individual cases with a local convergence criterion, the latter being what we'd referred to as "by grid point".
   - We have changed 'by grid points' to 'case-by-case' on line 135 of the revised manuscript, and on line 20-21 (in the abstract).

10. Line 140: "HETP does not consider the reverse problem…" Please provide an explanation why not? Is that case irrelevant in the applications for which HETP is designed – or for another reason?
    - We have included the following justification for not considering the reverse mode in this work.
    - Lines 158-164: "**For measured data, the reverse problem is typically not recommended since it lacks the inclusion of gas phase speciation in the input, making its predictions highly sensitive to measurement errors. For example, Hennigan et al., (2015) show that a ±10 % measurement error in $NH_4^+$ can alter the pH predicted by the reverse mode by more than 1 pH unit. Furthermore, Song et al., (2018) found that the aerosol pH predicted by the reverse mode may result in a bimodal pH distribution; in their study a negative ion balance gave highly acidic conditions while a positive ion balance gave near-neutral conditions. We note that the reverse mode is used in CMAQ to perform mass transfer with the coarse mode (Pye et al., 2022), but other CTMs that employ ISORROPIA use only the forward mode.**"

11. Line 149: "expressed as molar equivalent Na, …" For those non-volatile cations, I guess this should be stated as $Na^+$, $K^+$, $Mg^{2+}$ etc. – or is the charge omitted for a reason?
    - Thank you for pointing this out. This has been corrected in the manuscript.
    - Lines 171-173 now read: "… total sodium (TNa, expressed as molar equivalent **$Na^+$**), total chloride (TCl, expressed as molar equivalent HCl), total magnesium (TMg, expressed as molar equivalent **$Mg^{2+}$**), total potassium (TK, expressed as molar equivalent **$K^+$**) and total calcium (TCa, expressed as molar equivalent **$Ca^{2+}$**)."

12. Line 156: "As a result, …" This seems not to be a "result" but rather a choice or consideration, right?
    - The wording has been modified. Line 183 has been updated to: "**Considering these** mass violations…".

13. Line 168: "mean activity coefficients"; Are these molality-based mean activity coefficients or molarity-based or mole-fraction-based? It should be mentioned somewhere. In the following line, "non-ideal solutions" should be non-ideal aqueous solutions because water is used as reference solvent for the ions.
    - Line 194-196 have been modified to read: "ER1 to ER7 are solved by introducing additional relationships for mass conservation, electroneutrality (i.e., a charge balance equation), aerosol water activity, and mean **molality-based** activity coefficients ($\gamma$) to represent ion–ion interactions in non–ideal **aqueous** solutions ($\gamma \to 1$ as the solution becomes more dilute, i.e. more "ideal")."

14. Lines 185–188: In this context, do the free amounts of present components mean that charge balance is violated. If charge balance is violated, is additional $H^+$ or $OH^-$ considered to balance the charges and allow for all "free" amounts to be accounted for in aqueous solution?
    - The free amounts are not included in the charge balance of the system, and hence yes, the charge balance is violated when the free amounts are included. The free amounts may be associated with weaker anions not considered in the current system of equations (e.g., bicarbonate and carbonate ions ($HCO_3^-$, $CO_3^{2-}$) which may result from the dissociation of calcium carbonate or the uptake of carbon dioxide into water). Hence, we neglect the impact weaker anions on the $H^+$ concentration, which is assumed to be minor (the $H^+$ concentration is controlled by the stronger anions and cations). The stronger anions considered in the system are charge-balanced with the available cations.

15. Line 191: "… to prevent loss of mass of species such as Na, Mg, K, and Ca; the free mass must be added back to the captured mass partitioned by the solver prior to returning to the program accessing the inorganic heterogeneous chemistry solver." When such cases occur, does the presence of free mass not indicate a problem in the inputs? In other words, if the inputs are not charge balanced, and presumably ions like H+ and OH- not directly measured or traced, would ignoring such free mass really be an important issue or rather reflect minor problems with the accuracy and consistency of input species amounts? Please discuss briefly.

   - Ignoring the free amounts upon output would be an important issue in a mass-conserving model, but only if the free amounts are lost in the solver call when the output is transferred back to the model. We note that state-of-the-art air-quality models (i.e., CMAQ, GEOS-Chem) do not lose this mass since only the semi-volatile partitioning species ($Cl^-$–HCl, $NO_3^-$–$HNO_3$, $NH_4^+$–$NH_3$) are modified upon output from the solver (see also the response to #16 below). The adjustments at the initialization of HETP and ISORROPIA (that generate free amounts) are performed to avoid excess particle alkalinity that exceeds dissolved particulate calcium carbonate (Pye et al., 2020). Therefore, these initial adjustments are done to ensure the resulting equilibrium solution is "atmospherically relevant".

     Lines 214-219 in the revised manuscript have been updated to: **"In addition to the free amounts generated within a chemical subspace during dry salt partitioning, free amounts may also be generated during the initialization of HETP and ISORROPIA, prior to entering a chemical subspace. Specifically, automatic adjustments are applied if the input precursor species are nonelectroneutral. In this case, any excess cations are ignored, and free amounts of Na, Ca, K and Mg may be created. These automatic adjustments help constrain the particle alkalinity of the equilibrium solution, ensuring that it does not exceed the pH of dissolved particulate calcium carbonate (Pye et al., 2020)."**

16. Line 198: related to previous comment; "these implementations may be inadvertently losing aerosol mass due to this issue…"; but that loss is presumably only happening at initialization, since afterwards all species should be part of the "captured" portion, or not? Please comment.

   - Thank you very much for this comment.
   - Investigating the CMAQ code in more detail in response to the referee's comment, we have found that the mass of inorganic non-volatile species (Na, Ca, K, Mg) is *not* being lost in the model. The output from ISORROPIA for $Na^+$, $Ca^{2+}$, $K^+$ and $Mg^{2+}$ are not used to update the non-volatile species outside of the routine calling ISORROPIA II. This can be seen by backtracking the logic of CMAQ before and after the ISORROPIA II call. For the fine mode, CMAQ executes ISORROPIA in the forward metastable mode (subroutine 'VOLINORG'). Upon output from the solver, only the inorganic gases ($NH_3$, $HNO_3$ and HCl) are exported to the calling code. Thus, the non-volatile species are only considered at input during the call to ISORROPIA and are not modified upon output from the solver. Therefore, the net mass of the non-volatile species before and after the call to ISORROPIA does not change in CMAQ (any free mass created for non-volatile species is not allowed to affect the net mass of these species outside of the ISORROPIA solver). Hence, in CMAQ the aerosol mass is conserved during the call to ISORROPIA. Likewise in GEOS-Chem, where the non-volatile species are not used at output from the solver and do not change as a result of the ISORROPIA call.
   - Lines 224-230 in the revised manuscript has been modified to reflect the above: **"If the non-volatile species ($Ca^{2+}$, $Mg^{2+}$, $K^+$, $Na^+$) output by the solver are used by the CTM, and the 'free' amounts are not retained and used to conserve mass, then inputs to the solver which result in 'free' species will be lost in the solver call. We note, however, that CTMs such as CMAQv5.4 and GEOS–Chem v14.0.2 avoid this potential problem by only allowing the semi-volatile species (i.e., $Cl^-$–HCl, $NO_3^-$–$HNO_3$, $NH_4^+$–$NH_3$) to be modified on output from the solver. The semi-volatile species are then saved and transferred back to the model. The non-volatile species are not used after chemical partitioning and are not transferred back to the model calling ISORROPIA. Therefore, any non-volatile free mass that was created in ISORORPIA is not lost in the solver call in these CTMs (aerosol mass is conserved)."**

17. Table 1: ER1 row on right, please check the notation for the activity coefficients. It is unclear what gamma_H2SO4 means compared to gamma_H-HSO4.

   - ER1 in Table 1 has been modified to $\frac{\gamma_{H^+}\gamma_{SO_4^{2-}}}{\gamma_{HSO_4^-}}$ in the revised manuscript.
   - The following details have been added as a footnote to the table:

     "*Note: $\frac{\gamma_{H^+}\gamma_{SO_4^{2-}}}{\gamma_{HSO_4^-}} = \frac{\gamma_{H^+}^2\gamma_{SO_4^{2-}}}{\gamma_{H^+}\gamma_{HSO_4^-}} = \frac{\gamma_{H_2SO_4}^3}{\gamma_{H-HSO_4}^2}$ (Kim and Seinfeld, 1993b)"

Furthermore, presumably the [H+] notation refers to molalities of H+, not molarity, please clarify in the caption? Also, actual molalities should be normalized by unit molality in such equilibrium relationships, which means that any equilibrium constant is a dimensionless quantity, but scale-dependent; see e.g. Pye et al. (2020). The often-stated units of the equilibrium constants arise from a less rigorous (but common) use of dropping any unit molalities and unit pressure values. I think this should be mentioned in a footnote to this table.

- The table caption of the revised manuscript has the following text:
    I. Lines 238-239: "**The ions denoted in square brackets […] (i.e., [H$^+$], [SO$_4^{2-}$], [HSO$_4^-$], etc.) refer to molalities with units of mol kg$^{-1}$.**"
    II. Lines 240-241: "**Theoretically, equilibrium constants are unitless since each pressure or concentration should be normalized by a standard state; here standard states are neglected.**"

18. Lines 218 and 223: (kJ mol^-1); Shouldn't these stated units be (J mol^-1) to remain consistent with the notation and values used in Table 1? While such values are typically tabulated in kJ/mol, in the application here they are properly converted to J/mol and that is the SI unit that I'd recommend being stated.
    - Thank you very much for pointing this out. The units have been corrected to J mol$^{-1}$ in the revised manuscript.

19. Line 230–231: "correlation" should be "relation". And, clarify "where the water activity ($a_w$) is equal to the fractional relative humidity (0 to 1 scale)." This sentence could be improved. The ZSR relation is the assumption that the water uptake by individual components of a mixture equilibrated at a constant RH is assumed to be additive. It does not state that a_w is equal to RH, which is a statement of vapor-liquid equilibrium (modified Raoult's law) for bulk solutions.
    - Thank you for pointing out the poor description in the initial manuscript.
    - We have updated the text in the revised manuscript on lines 262-271:
    "**Aerosol liquid water content in kg m$^{-3}$ air is calculated according to the Zdanovskii−Stokes−Robinson (ZSR) relation (Robinson and Stokes, 1965), as**
    $$W = \sum_i \frac{M_i}{m_i(a_w)},$$ **(3)**
    **where $M_i$ is the concentration of species $i$ in mol m$^{-3}$ air and $m_i$ is the molality (mol kg$^{-1}$) of an aqueous solution of $i$ at the same water activity ($a_w$) as the mixture. It is assumed that there are negligible effects from droplet curvature (i.e., Kelvin effect), and that the growth of an aerosol by uptake of H$_2$O does not affect the ambient water vapor pressure (i.e., no effect on the ambient relative humidity). Therefore, equilibrium between the vapor (gas) and liquid (aerosol) phase is assumed, with $a_w$ = RH. (Seinfeld and Pandis, 2016)**"

20. Figure 1, caption text: "subcases H6, G5, M8, …". Please point out that subcase H6 refers to CALCH6 in the figure; this wasn't obvious on a first read.
    - Thank you for noting this as a potential source of confusion.
    - This has been modified in the revised manuscript, and 'CALC' has been added before the subcases in caption text of Figure 1: "… subcases **CALC**H6, **CALC**G5, **CALC**M8, **CALC**O7 and **CALC**P13…."

21. Line 278: "ITP has the advantage of 'superlinear convergence', and hence …". So do Brent's method or Ridder's method, e.g., outlined in Numerical Recipes (https://search.worldcat.org/search?q=bn:0521880688). Have the authors considered such well-established alternatives?
    - Oliveira and Takahasi (2021) compared the performance of the interpolate, truncate and project (ITP) algorithm against other well-established root-finding methods (i.e., Ridder's method, the Illinois method, Matlab's fzero routine and the Secant method). They tested 24 different mathematical functions and tabulated the number of function evaluations required to achieve convergence (to a set tolerance) for each root-finding method. When compared against Ridder's method, for example, ITP required on average 20.2 function evaluations while Ridder's method required 26.1 (about 30% more function evaluations than ITP). Out of all the mathematical functions investigated, the worst-case performance for ITP was 34 function evaluations, while for Ridder's method it was 68. Oliveira and Takahasi (2021) also point out that even the original unmodified bisection method outperforms Ridder's method in about 25% of the functions investigated.
    - We have included the following text on lines 125-130 in the introduction of the revised manuscript:
    "**Oliveira and Takahasi (2021) also compared the performance of ITP against well-established alternative root-finding methods, such as Ridder's method, the Illinois method, Matlab's 'fzero' routine and the Secant method. For all functions evaluated for convergence, ITP**

required the least amount of function evaluations, when compared against the other root-finding methods. For example, compared to Ridder's method, ITP requires an average of 20.2 function evaluations while Ridder's method requires an average of 26.1."

22. Line 299: "and hence $x$ is not an accurate solution to the system of equations." … at the targeted tolerance level for x (I assume). Could it be in some cases that the accepted tolerance on x is simply too loose to determine an adequate |f(x)| ~= 0.0 value (a problem with numerical computations) even if the function is continuous?
   - This is a good point, and we have updated the revised manuscript to reflect this possibility.
   - Lines 337-340 have been modified in the revised manuscript: "This oscillating behavior of $y$ may indicate (i) that $x$ is a discontinuity, (ii) that there is significant non-linearity in the partitioning solution, **or (iii) that the accepted tolerance on $x$ is too loose for convergence**, and hence $x$ is not an accurate solution to the system of equations **at the targeted tolerance level for $x$.**"

23. Lines 308–313: Regarding solving quadratic equations with the standard formula, this is a known problem; another option is to use the method outlined in Section 5.6 of the Numerical Recipes book.
   - Thank you very much for pointing out this analytic formula to solve the quadratic formula. We were previously not aware of this alternative method.
   - We have implemented the analytic formula to solve the quadratic equation (given in the Numerical Recipes book by Press et al., (2007)) into the HETP code. The resulting changes to the output are minor (i.e., there are no visual changes to the figures in the manuscript). However, for some input conditions, using this alternative (exact) formula to solve the quadratic equation produces an equilibrium solution with lower error statistics. Given the positive performance of this alternative formula, we have set it as the default option in the HETP code. The Taylor expansion remains in the code, but it is now commented out.
   - As a result of updating the methodology to solve quadratic equations in the HETP code, some values in Table 2, Table 6, Table S4 and Table S5 (i.e., error statistics) of the revised manuscript have improved.
   - Lines 347-362 in the revised manuscript have been updated: "Quadratic equations have the form $f(x) = ax^2 + bx + c$, where the solution corresponding to $f(x) = 0$, is usually expressed as the **standard** quadratic formula $x = \frac{-b \pm \sqrt{b^2 - 4ac}}{2a}$. $x$ **has two possible solutions, $x_1$ and $x_2$, determined by the sign in front of the radical**. As identified in Makar et al. (2003) in the original version of HETV, when the coefficient '$b$' differs by several orders of magnitude from coefficients '$a$' or '$c$', floating-point arithmetic can fail to give an accurate answer for $x$ when using the standard root formula. For example, if $\sqrt{b^2 - 4ac} \approx b$, then addition in the quadratic formula may be problematic since we are subtracting two nearly equal numbers (i.e., $\approx -b + b$). **To avoid this issue, HETP uses the analytic formula given in Press et al., (2007) to solve the quadratic equation: $q = -\frac{1}{2}\left(b + sign(b)\sqrt{b^2 - 4ac}\right)$ with roots $x_{p_1} = \frac{c}{q}$ and $x_{p_2} = \frac{q}{a}$. Care must be taken when applying this formula since the appropriate choice of $x_{p_1}$ and $x_{p_2}$ depends simultaneously on the chosen solution (i.e., either $x_1$ or $x_2$) and the sign of the $b$ coefficient, as described in Table S3 of the supplement. In addition to analytic formula from Press et al., (2007), HETP also includes code (which is commented out) to solve the quadratic equation using a Taylor series expansion of the quadratic formula. In this code, the Taylor series expansion is only applied when the coefficients '$b$' and '$c$' differ by orders of magnitude, and hence when the numerical precision issues as described above are likely to occur (note that $a = 1$ in all subroutines; formulae were normalized). Both methods produce very similar results, but the analytic formula provided by Press et al., (2007) is superior to the Taylor expansion since it provides an exact solution, giving lower error metrics (i.e., Sect. 4).**"

**Table S3:** The analytic solution of the quadratic formula used in HETP, as given by Press et al. (2007).

| Assume a quadratic equation of the form $ax^2 + bx + c = 0$. In ISORROPIA II the quadratic equation is solved using the unmodified quadratic formula, with two possible roots | In HETP we have updated the code so that it no longer uses (1) and (2) to solve the quadratic equation. Instead HETP uses (3), with two possible cases as shown below: |
|---|---|

Left column:

$$x_a = \frac{-b+\sqrt{b^2-4ac}}{2a}, \qquad (1)$$

and

$$x_b = \frac{-b-\sqrt{b^2-4ac}}{2a}. \qquad (2)$$

Press et al. (2007) gives an analytic formula to solve the quadratic equation that avoids the catastrophic cancellation experienced in (1) and (2),

$$q = -\frac{1}{2}\left(b + sign(b)\sqrt{b^2-4ac}\right) \text{ with } x_1 = \frac{c}{q} \text{ and } x_2 = \frac{q}{a}. \qquad (3)$$

Expanding $x_1$

$$x_1 = \frac{c}{q} = \frac{-2c}{b + sign(b)\sqrt{b^2-4ac}}$$

and rationalizing the denominator gives*

$$x_1 = \frac{-2c}{b+sign(b)\sqrt{b^2-4ac}} \times \frac{b-sign(b)\sqrt{b^2-4ac}}{b-sign(b)\sqrt{b^2-4ac}} = \frac{-b+sign(b)\sqrt{b^2-4ac}}{2a}. \qquad (4)$$

*(4) **is not** used in the HETP code. In the HETP code, $x_1 = \frac{c}{q}$, as given in (3).

Likewise, expanding $x_2$ gives

$$x_2 = \frac{q}{a} = \frac{-b-sign(b)\sqrt{b^2-4ac}}{2a}. \qquad (5)$$

Right column:

**Case 1: The solution is (1).**
```
if (b > 0) then
```
$$x = x_1 = \frac{c}{q} = \frac{-b+sign(b)\sqrt{b^2-4ac}}{2a} \equiv \frac{-b+\sqrt{b^2-4ac}}{2a}$$
```
elseif (b < 0) then
```
$$x = x_2 = \frac{q}{a} = \frac{-b-sign(b)\sqrt{b^2-4ac}}{2a} \equiv \frac{-b+\sqrt{b^2-4ac}}{2a}$$
```
elseif (b = 0) then
```
$$x = \frac{\sqrt{-4ac}}{2a} \text{ if } ac < 0, \text{ otherwise undefined}$$
```
end if
```

**Case 2: The solution is (2).**
```
if (b < 0) then
```
$$x = x_1 = \frac{c}{q} = \frac{-b+sign(b)\sqrt{b^2-4ac}}{2a} \equiv \frac{-b-\sqrt{b^2-4ac}}{2a}$$
```
elseif (b > 0) then
```
$$x = x_2 = \frac{q}{a} = \frac{-b-sign(b)\sqrt{b^2-4ac}}{2a} \equiv \frac{-b-\sqrt{b^2-4ac}}{2a}$$
```
elseif (b = 0) then
```
$$x = \frac{-\sqrt{-4ac}}{2a} \text{ if } ac > 0, \text{ otherwise undefined}$$
```
end if
```

In some instances, the sign of $b$ is known at compile time (i.e., $b$ always $> 0$), eliminating the need for a branching if statement.

24. Line 355: "Removing function and subroutine calls, except for process calls …". Unclear; do you mean that a lot of the subroutine contents have been merged into a single subroutine? That would seem to be a disadvantage in code readability and unnecessary given that modern Fortran allows for assumed-shape array association via subroutine and function interfaces that avoid the generation of temporary local copies... Please clarify this a bit more.
   - Thank you for this comment.
   - HETP has merged some subroutines into a single file, such as the calculation of the aerosol liquid water content and evaluating the system of equations (which is done within a function in ISORROPIA II). This allows several static variables to be calculated once at the start of the subroutine and then reused as needed during the iterative process and avoids generating a hard-to-follow function or subroutine that contains an excessive number of arguments. In ISORROPIA II, these static variables are continually recalculated, thereby unnecessarily increasing the execution time relative to HETP.
   - However, considering your comments on code readability, we have decided to create 5 new subroutines that remove code sections identically repeated in several different subroutines. Specifically, these are subroutines that solve the minor systems and calculate the [H+] concentration (calchso4, calchno3, calcnh3, calchclhno3, calcph).
   - Lines 405-408 have been updated in the revised manuscript: "Removing function and **short** subroutine calls, except for process calls to calculate activity coefficients (calcact), to solve a cubic equation (poly3), **to solve minor systems**, and to perform a post-convergence mass balance adjustment (adjust). **The merging of functions and some short subroutines allowed several variables to be calculated once and reused throughout the iterative process, reducing computational time**,"

25. Line 376: "… complied using the '-r8' flag…" Spelling of compiled; also clarify which compiler is used with this flag, since it may differ amongst various options. Also, if ISORROPIA is not used with double precision floating point numbers in its usual implementation within models like GEOS-Chem, would you expect the move from single to double precision to slow down ISORROPIA substantially?
   - 'complied' has been updated to 'compiled' on line 427 of the revised manuscript.
   - The complier version has been added to the text on lines 435: "**The compiler used was an intel compiler (IFORT) version 2021.5.0.2021109.**"
   - We did not find a noticeable difference in the execution time of ISORROPIA when it is complied with the '-r8' flag versus when it is not.
   - Lines 428-432 of the revised manuscript have been updated: "**It should be noted that ISORROPIA is coded to use mostly 'double precision' variables, but some single precision variables exist (i.e., declared as 'real', either explicitly or by default under Fortan variable**

**naming conventions). While compiling ISORROPIA with the '-r8' flag does not have a large impact on the execution time, it may in some cases produce non-trivial differences in the output, compared to output produced without the '-r8' flag.**"

26. Related to above, other compiler flags, such as for fast math or a certain optimization level may also matter a lot in terms of run time performance; were those consistent between ISORROPIA and HETP compilations? It would be good to state somewhere in the SI the compiler flags used for the model comparison runs.
    - We did not use any special complier flags when compiling HETP. For ISORROPIA II the only flag used during compilation was '-r8'. We avoided the use of optimization flags when comparing HETP against ISORROPIA II, since the GEOS-Chem group has reported reproducibility issues with ISORROPIA II when such flags are applied during compilation. For example, on the GEOS-Chem Harvard University Wiki Page (http://wiki.seas.harvard.edu/geos-chem/index.php/ISORROPIA_II) Shannon Capps wrote the following regarding ISORROPIA II compiler optimizations: "The -fp-source option of the Intel Fortran Compiler tells the compiler to only do "safe" optimizations (i.e. nothing that would affect the numerical precision of the output). For some reason, ISORROPIA II is very sensitive to the compiler optimization."
    - We have included the following text on line 432-433 of the revised manuscript: "**Aside from the '-r8' flag, no other compilation flags were used in this work.**"

27. Figure 4, Line 560: "The air temperature and relative humidity are 243 K and 5% respectively,".
    Was this combination of temperature and RH used because the demonstrated numerical issues only surface under such thermodynamic conditions? At least in the lower to mid troposphere, such RH and temperature conditions seem rare and when present, how likely would it be that crystallization of certain salts could safely be ignored at 5 % RH? In other words, while the demonstration is good, one might wonder whether these issues within ISORROPIA matter in applications within air quality models – and whether the solution shown with HETP in panel (d) would represent a realistic case or a very hypothetical strongly supersaturated metastable state.
    - While 243 K (-30 ºC) seems low, this is a relatively frequent surface air temperature in Canada during the winter season, as well as in the lower to mid troposphere. While such low air temperatures are likely to coexist with a relative humidity (RH) > 35 % at the surface, we chose a RH of 5 % for this demonstration to highlight the numerical issues present in ISORROPIA II, which are more frequent and pronounced at such a low RH. The numerical issues highlighted in Fig. 4 continue to be present (albeit to a lesser extent) in the output from ISORROPIA even as the relative humidity is increased for the same set of initial conditions. Specifically, ISORROPIA will continue to output negative $HSO_4^-$ until a RH of about 25 %, and the discontinuity in $NO_3^-$ and $Cl^-$ is still present at a RH of 35 %.
    - The issue of low RH may be more pronounced at the surface in hot arid climates where a RH < 10 % is likely to be rather common. For example, Shaw and Rood (1990) measured ambient aerosols in California, where a RH < 10 % was not uncommon. Furthermore, they measured aqueous aerosols existing in a metastable state at a RH as low as 4%, demonstrating that even at such a low RH an aqueous phase is still possible. In fact, about 2% of their measurements had crystallization of aerosols at a RH < 10 %.
    - Nonetheless, we agree with the referee that the atmospheric conditions considered here (T = 243 K and RH = 5 %) are likely to represent a very hypothetical case.
    - The following text has been added to lines 568-578 of the revised manuscript: "**While such a combination of air temperature and relative humidity is likely to be rare in the lower troposphere, it is not uncommon for surface air temperatures to reach 243 K or lower in the winter in Canada, and at similar higher latitudes in other parts of the world. The choice of RH = 5 % here is used to highlight the numerical issues present in ISORROPIA, which occur more frequently and are more pronounced at low RH. However, the numerical issues highlighted in Fig. 4 continue to be present in the output from ISORROPIA, but to a lesser extent, even as the RH is increased to 35 % for the same set of initial conditions. At an ambient RH of 5 % the assumption of a supersaturated aqueous phase may be less justified and is more likely to be representative of a very hypothetical case. Nonetheless, observations from southern California have indicated (although in warmer air temperatures than investigated here) that crystallization of some ambient aerosols may not occur until a RH as low as 4 % (Shaw and Rood, 1990), suggesting that in some atmospheric conditions metastable aerosols are possible even at a very low RH of 5 %.**"

28. Figure 5: The case shown in panel (d) seems to matter most in terms of numerical inaccuracies, while panels (a)–(c) show excellent to sufficient agreement. Slight disagreements in panel (c) only become notable for very low molar amounts, which probably do not matter from an air quality point of view, would you agree?

- We agree and have noted this on line 651-653 of the revised manuscript: "**The differences between the two solvers noted for Cl⁻ and HCl in Fig. 5(c) are only for very low concentrations, which likely would not be impactful in practical air-quality applications.**"

29. Figure 6: Looking at the relatively poor performance of both models for the H6 category: are these just indicative of an attempt to solve equations for which double precision arithmetic is insufficient? If one were to set a threshold for accepted convergence, say of 10^-3 in terms of the xi' metric, how many out of all computations would violate this? Such an implications-oriented quantification (thinking of getting the inorganic aerosol mass concentration and composition roughly right) may be of interest for flagging the fraction of insufficiently solved HETP or ISORROPIA predictions in 3D models.

- **\* Equations in our response below assume unity activity coefficients for simplicity of presentation – we note the same reasoning applies for non-ideal solutions.**

  We feel that the issue here is not one of precision, but rather the manner in which the H⁺ ion is treated within the original ISORROPIA II CALCH6 algorithm, versus our use of $K_{HCl}$ and $K_{HNO_3}$ as metrics for model performance in Figure 6. Algorithm H6 proceeds (see SI, page 8) by searching for a root within a set interval. The 'objective function' used to define this root is obtained by combining different equilibrium expressions together and eliminating H⁺, giving

$$\frac{[NH_4^+][Cl^-]}{(RT)^2[HCl][NH_3]K_{HCl}\frac{K_{NH_3}}{K_{H_2O}}} - 1 = 0. \qquad (1)$$

  Note that Eq. (1), which defines the root, no longer includes an explicit requirement of convergence on the H⁺ concentration (or NO₃⁻, which is also a part of the system of equations). Instead, the determination of the root focuses on the remaining ions and gases in the system. CALCH6 is unique in this regard since it is the only chemical subspace that does not include H⁺ in the objective function when root-finding is necessary. The two equilibrium equations for $K_{HCl}$ and $K_{HNO_3}$ used in our comparisons (i.e., from Table 1), however, make use of the H⁺ concentration explicitly:

$$\frac{[H^+][Cl^-]}{RT[HCl]} - K_{HCl} = \xi \qquad (2)$$

  and

$$\frac{[H^+][NO_3^-]}{RT[HNO_3]} - K_{HNO_3} = \xi. \qquad (3)$$

  Equations (2) and (3) used in Fig. 6 explicitly evaluate the convergence of the *hydrogen ion* relative to the $K_{HCl}$ and $K_{HNO_3}$ equilibria, in addition to the convergence of the chloride and nitrate ions and gases. The relatively poor performance of the H6 algorithm thus tells us that although the other terms in the H6 chemical subspace have converged with the existing solution procedure (Eq. 1), convergence with respect to H⁺ remains poor (and possibly NO₃⁻). One potential means to remedy the situation may be to include a second convergence criteria for H6 based on additional equilibrium relationships that explicitly include H⁺ (and NO₃⁻) to ensure that it has converged equally well for all species considered in the chemical subspace. We note that this accuracy improvement (for either HETP or ISORROPIA II) might come with a performance cost, due to additional iterations being potentially required to achieve convergence for the hydrogen, nitrate, ammonium, and chloride ions.

- We have included the following text on lines 674-679 of the revised manuscript: "**H6 is unique relative to the other subspaces requiring a root-finding method (i.e., G5, O7, M8 and P13), since the objective function used to determine the root of the system of equations does not include H⁺ explicitly. The expressions for $\xi'$ used in Fig. 6, however, explicitly evaluate the convergence of H⁺ relative to $K_{HCl}$ and $K_{HNO_3}$ equilibria. The relatively poor performance of the H6 algorithm when evaluated using $\xi'$ thus tells us that although the other ions and gases in the H6 chemical subspace have converged with the existing solution procedure, convergence with respect to H⁺ remains poor.**"

**Comments on supplement and code:**

30. HETP package code: I had a look at the Fortran source code and have some related minor comments:

    a. Instead of using fixed kinds for reals and integers (like kind=8), it is better for clarity and portability to use a module that sets kind parameters, e.g. often something like a "dp" or "wp" kind for double precision (64-bit floating point numbers); a simple example would be: integer, parameter, public :: wp = kind(0.0D0) and use that parameter to declare all real(wp) variables. This will work with any compiler and is thus better than using kind=8 (which does not exist in all Fortran compilers, some using a different kind numbering system). Related, literal values would then be written e.g. as tstd = 298.15_wp instead of 298.15d0 and 82.0567d-6 would be 82.0567E-6_wp. Similar for integers and logicals (one could simply use default integer and logical (no specific kind stated). See https://fortran-lang.org/learn/best_practices/floating_point/ .

    - This has been updated throughout the code. In the module mach_hetp_mod.ftn90 the kind is now declared as "integer, parameter :: dp = kind(0.0d0)", and _dp has been used in place of d0. Additionally, all (kind=8) have been replaced with (kind=dp), and integers and logical variables use the default setting.

    b. In module mach_hetp_mod, instead of using the "save" attribute module-wide, it would seem unnecessary if the various water activity parameters, such as awsc and awss were defined as parameter arrays; e.g. real(wp), parameter :: awsc(100) = real( [28.16, 28.16, 28.16, …, 0.1], kind=wp)

    - The "save" attribute has been removed, and the arrays have been declared as parameters, as suggested by the referee.

    c. It seems that most subroutines are written as quasi-external, standalone procedures provided in the same file. Ideally, with modern Fortran, one would place them inside of one or several modules, so that their interfaces are implicitly known within the module's use scope and assumed-shape arrays could be used. Many of the subroutines may also qualify as "pure subroutines".

    - We have followed the approach of CC#1 and their implementation into GEOS-Chem:

        I. The code for all HETP subroutines has been placed within a module file (hetp_mod.ftn90), with all subroutines declared as private, except mach_hetp_15cases.

        II. Another module file (mach_hetp_mod.ftn90) holds the water activity arrays, kind declarations and convergence variables.

    d. Subroutine mach_hetp_calcd3 includes logical variable statement evaluations like "if (soln == .false. .and. frst == .true.) then". This is non-standard Fortran; the correct, standard way of writing the same would be: "if ((.not. soln) .and. frst) then".

    - All logical variable statement evaluations have been updated to meet the Fortran standard throughout the code.

**New references**

Amundson, N. R., Caboussat, A., He, J. W., Martynenko, A. V., Savarin, V. B., Seinfeld, J. H., and Yoo, K. Y.: A new inorganic atmospheric aerosol phase equilibrium model (UHAERO), Atmospheric Chemistry and Physics, 6(4), 975–992, https://doi.org/10.5194/acp-6-975-2006, 2006.

Capps, S. L., Henze, D. K., Hakami, A., Russell, A. G., & Nenes, A.: ANISORROPIA: The adjoint of the aerosol thermodynamic model ISORROPIA, Atmospheric Chemistry and Physics, 12(1), 527–543, https://doi.org/10.5194/acp-12-527-2012, 2012.

Hennigan, C. J., Izumi, J., Sullivan, A. P., Weber, R. J., and Nenes, A.: A critical evaluation of proxy methods used to estimate the acidity of atmospheric particles, Atmospheric Chemistry and Physics, 15(5), 2775–2790, https://doi.org/10.5194/acp-15-2775-2015, 2015.

Press, W. H., Teukolsky, S. A., Vetterling, W. T., Flannery B. P.,: Numerical Recipes The Art of Scientific Computing (3rd ed.), Cambridge University Press, Cambridge, UK, 1235 pp., ISBN 978-0-511-33555-6, 2007.

Pye, H. O., Nenes, A., Alexander, B., Ault, A. P., Barth, M. C., Clegg, S. L., Collett Jr., J. L., Fahey, K. M., Hennigan, C. J., Herrmann, H., Kanakidou, M., Kelly, J. T., Ku, I.-T., McNeill, V. F., Riemer, N., Schaefer, T., Shi, G., Tilgner, A., Walker, J. T., [...] and Zuend, A.: The acidity of atmospheric particles and clouds, Atmospheric Chemistry and Physics, 20(8), 4809–4888, https://doi.org/10.5194/acp-20-4809-2020, 2020.

Saiz-Lopez, A., Plane, J. M., Baker, A. R., Carpenter, L. J., von Glasow, R., Gómez Martín, J. C., McFiggans, G., & Saunders, R. W.: Atmospheric chemistry of iodine, Chemical Reviews, 112(3), 1773–1804, https://doi.org/10.1021/cr200029u, 2011.

Seinfeld, J. H., and Pandis, S. N.: Atmospheric Chemistry and physics: From air pollution to climate change, Wiley & Sons, 2016.

Shaw, M. A., and Rood, M. J.: Measurement of the crystallization humidities of ambient aerosol particles, Atmospheric Environment, Part A. General Topics, 24(7), 1837–1841, https://doi.org/10.1016/0960-1686(90)90516-p, 1990.

Song, S., Gao, M., Xu, W., Shao, J., Shi, G., Wang, S., Wang, Y., Sun, Y., and McElroy, M. B.: Fine-particle ph for Beijing winter haze as inferred from different thermodynamic equilibrium models, Atmospheric Chemistry and Physics, 18(10), 7423–7438, https://doi.org/10.5194/acp-18-7423-2018, 2018.

**Referee #2**

In this work the authors present their HETP inorganic thermodynamic partitioning solver, an evolution of the ISORROPIA algorithms commonly embedded in many current atmospheric chemistry models. Based on the stability, accuracy, and efficiency metrics presented here, this solver is an important development that should see rapid adoption within the atmospheric modeling community. On the whole I find this manuscript to be very well planned and composed, with clear figures and text. I have just a few suggestions on how to strengthen an otherwise excellent paper, and I recommend publication following just a few minor changes.

- We would like to thank Referee #2 for taking the time to provide valuable feedback to help strengthen our work. We appreciate the positive feedback provided above.

1. Formatting of "GEOS-Chem" should be fixed to remove all-caps from the final "hem".
   - Thank you for noticing this. We have updated the revised manuscript and changed 'CHEM' to 'Chem' throughout.

2. Lines 155-156 and other similar references mention mass conservation issues in ISORROPIA. Obviously mass conservation is an important modeling goal for a number of reasons, but it would be helpful to have some context on the relative and absolute scale of these violations. How much mass are we talking about here, and how impactful might it be in common practice?
   - This is a very good suggestion to include in the manuscript. We have quantified mass conservation violations in two places in the revised manuscript to highlight the significance of this issue. The bolded text below gives the specific additions included in the revised manuscript, but we have included some earlier text to help provide context to these additions.
     - Lines 179-183 (bolded text) of the revised manuscript: "For some input conditions ISORROPIA will adjust the input precursor concentrations prior to determining the subroutine that should be entered. Specifically, ISORROPIA will adjust TA and TCl so that they are no less than $1\times10^{-10}$ mol m$^{-3}$, and if (TNa + TS + TN) < $1\times10^{-10}$ mol m$^{-3}$, then ISORROPIA will adjust TNa and TN so that they are no less than $1\times10^{-10}$ mol m$^{-3}$ (note these are applicable only to Branch 3 and 4; see Fig. 1). These adjustments performed within a chemical transport model result in output speciation that violates mass conservation, since mass is created for TA, TN, TCl and TNa. **For example, for 50,000 unique sets of input executing Branch 4 subroutines (i.e., winter input from Sect. 4.2), performing these adjustments results in a median of $1.09\times10^{-3}$ ug m$^{-3}$ of TCl being created by the solver. On a relative scale $\left(\frac{\text{output mass}}{\text{input mass}} \times 100\ \%\right)$ this represents a median increase in TCl mass by 42.7 %; for 25 % of these input conditions the relative increase in TCl mass ≥ 4414 %. In an air-quality model these mass violations would occur at a single timestep, therefore the impact would increase as the simulation progresses.**"

     - Lines 545-548 (bolded text) of the revised manuscript: "The largest absolute differences of 100 % – 600 % are in L9, and are predominantly due to (ii), where for some input conditions ISORROPIA creates dry salt mass for TA, TS, and TK. **Specifically in ISORROPIA, 6.02 %, 0.05 % and 5.97 % of the test input conditions shown in Fig. 3(a-b) create mass for TS, TA, and TK respectively that cannot be attributed to machine precision near the lower limit used in the solver (i.e., $\text{Species}_{out} - \text{Species}_{in}$ > $9.999\times10^{-19}$ mol m$^{-3}$). The median relative mass created for these input conditions is 22.6 % for TS, 0.24 % for TA and $2.93\times10^{10}$ % for TK.**"

3. Figure 3: This figure could use some help to make it more legible and comprehensible. Label text sizes (especially axis labels) are unreasonably small relative to manuscript text. Also, it should be much easier to find and interpret the conditions that differentiate the top row (panels a-c) from the bottom row (panels d-f). I suggest that the values in lines 479-480 be relocated to a small table for easy reference, perhaps with some form of qualitative description for each (High/Low NaCaKMg?) for easier description.
  - We have updated Fig. 3 (shown below) to increase all text label sizes to make it more understandable.

[Figure]

  - We have added a table (Table 3) to the revised manuscript detailing the conditions of Fig. 3.
  - The following text and table have been added to lines 521-531 of the revised manuscript:
    "Figure 3 displays a comparison of HETP and ISORROPIA, where now TS and TA are varied simultaneously while all other input precursor species are held constant. **Figure 3 displays output generated from $n$ = 2,000,000 unique test cases. These cases are divided into two tests, denoted as a high $Mg^{2+}$-$Ca^{2+}$-$K^+$-$Na^+$ case (Fig. 3a-c) and a low $Mg^{2+}$-$Ca^{2+}$-$K^+$-$Na^+$ case (Fig. 3d-f). The input conditions used to generate Fig. 3 are summarized in Table 3.**"

  - **Table 3 –** Input conditions used to generate Fig 3 for 2,000,000 total unique test cases. All input precursor species have units of mol m$^{-3}$ air.

| | Fig(a-c): High $Mg^{2+}$-$Ca^{2+}$-$K^+$-$Na^+$ | Fig(d-f): Low $Mg^{2+}$-$Ca^{2+}$-$K^+$-$Na^+$ |
|---|---|---|
| **TS** | Varying between $2.5\times10^{-5}$ and $2.5\times10^{-12}$ | Varying between $2.5\times10^{-5}$ and $2.5\times10^{-12}$ |
| **TA** | Varying between $2.5\times10^{-5}$ and $2.5\times10^{-18}$ | Varying between $2.5\times10^{-5}$ and $2.5\times10^{-9}$ |
| **TN** | $3.0\times10^{-6}$ | $1\times10^{-8}$ |
| **TNa** | $1.0\times10^{-5}$ | $1\times10^{-6}$ |
| **TCl** | $1.0\times10^{-14}$ | $1\times10^{-14}$ |
| **TCa** | $1.0\times10^{-8}$ | $1\times10^{-16}$ |
| **TK** | $1.0\times10^{-14}$ | $1\times10^{-17}$ |
| **TMg** | $1.0\times10^{-14}$ | $1\times10^{-16}$ |
| **Temp (K)** | 306 | Same as Fig(a-c) |
| **RH (%)** | 35 | Same as Fig(a-c) |
| $n$ | 1,000,000 | 1,000,000 |

4. Figure 4: Panel titles may lead to confusion as written, as "HETP - base" could be interpreted as subtraction (i.e. a difference plot). I suggest switching to a colon ("HETP: base") for the HETP panels.
  - Thank you for pointing this out as a possible source of confusion. We have modified Fig. 4 to remove the dash, and replaced it with a colon, as suggested. We have applied this modification to other figures that also used a dash (Fig. 3, Fig. 5 and Fig. 6)

**Referee #3**

Review of "HETerogeneous vectorized or Parallel (HETPv1.0): An updated inorganic heterogeneous chemistry solver for metastable state NH4+–Na+–Ca2+–K+–Mg2+–SO42—NO3—Cl– based on ISORROPIA II " for Geoscientific Model Development

**General Comments**

Miller, Makar and Lee describe the development of a novel computational model written in Fortran 90 for the thermodynamic partitioning of a total amount of inorganic species as listed in the title between the gas and liquid phases in the presence of water. Typically, they follow the algorithms of the forward solution of ISORROPIA II for the metastable cases. Additionally, they identify and correct algorithmic formulations that lead to errors in the output of ISORROPIA. Furthermore, they implement a recently developed root finding approach instead of bisection that improves the accuracy of equilibrium solutions and the speed of arriving at them in most cases. The model is thoroughly tested not only against ISORROPIA but against the analytical solution of the equilibrium equations. Additionally, the accuracy and computational costs were explored for realistic cases as derived from the regional GEM-MACH model for selected conditions summer and winter. Finally, the code for this open source model has been made publicly available through Zenodo.

The manuscript is replete with evidence of the carefulness with which this model was developed and tested. The authors are clear as to where HETP outperforms ISORROPIA in terms of accuracy or computational cost but also where it underperforms, which is less common. In most cases, the authors posit fundamental explanations for why HETP performs differently than ISORROPIA, identifying specific algorithmic changes associated with the results in most tables and figures. For instance, the third subtable in Table 2 is evidence of an excellent investigation of the algorithmic explanation for the observed improvements. The authors have helpfully structured the manuscript by selecting tests that pertain to specific algorithmic changes to avoid inundating the reader with excessive comparisons while still demonstrating the extent of the testing conducted and then building to the comprehensive timing tests, which are sufficient to demonstrate the robustness of the model development process.

- We would like to thank Referee #3 for their careful review of our manuscript and the positive feedback provided above. We appreciate the comments this reviewer provided, particularly those aimed at improving the language and clarity of our work.

The authors are helpfully straightforward about HETP solving only the metastable cases. Since most chemical transport models only use the metastable solutions from ISORROPIA, this clarity is important but not impactful for those inclined to adopt the model. The primary change I think would improve the manuscript is to be clear at least in the abstract if not in the title that only the forward solution in ISORROPIA is included. Currently, the first mention that HETP only addresses the forward solution of ISORROPIA is at the beginning of Section 2 though in the introduction ISORROPIA is referenced as being used in CMAQ, in which both the forward and reverse solutions are employed for dynamic equilibrium of coarse mode aerosol with the rest of the population. So as to not cause a reader undue hope that well-documented issues with the reverse solution have been resolved with HETP, I would urge the authors to consider being clear that only the forward solution is implemented in HETP.

- We have included that HETP solves the forward mode of the ISORROPIA algorithms in the abstract, but not the title. We feel that since "forward" is a non-standard scientific term, its use in the title may lead to ambiguity, since it cannot be adequately defined.
- The sentence on lines 11-12 of the abstract in the revised manuscript has been updated: "Specifically, the code solves the system of equations describing the **"forward" (gas + aerosol input)** metastable state, but with algorithm improvements and corrections."
- We have also referenced that CMAQ solves the reverse mode in the updated manuscript (line 163-164).

With this small but important change, a few responses to specific comments aimed to add value for future readers, and a careful grammatical revision including reduction of the number of parenthetical phrases, I would expect that the publication of this manuscript would benefit many in the atmospheric modeling community for years to come.

- We have greatly reduced the number of parenthetical phrases throughout the revised manuscript.

**Specific Comments**

1. Lines 9–12 The claims in the first two lines of the introduction are very important and are well-supported in the manuscript. The current language in the first two sentences, especially the second, obscures the value of this paper somewhat. Consider rephrasing these two sentences, possibly into three shorter ones, for the sake of clarity and impact.
   - We have rephrased the two sentences into three shorter ones as suggested.
   - Lines 10-14 have been updated in the revised manuscript: "**We describe a new Fortran computer program to solve system of equations for the $NH_4^+$–$Na^+$–$Ca^{2+}$–$K^+$–$Mg^{2+}$–$SO_4^{2-}$–$NO_3^-$–$Cl^-$–$H_2O$ system, based on the algorithms of ISORROPIA II. Specifically, the code solves the system of equations describing the "forward" (gas + aerosol input) metastable state, but with algorithm improvements and corrections. These algorithm changes allow the code to deliver more accurate solution results in formal evaluations of accuracy of the roots of the systems of equations, while reducing processing time in practical applications by about 50 %.**"

2. Lines 80-92 The authors have reasonable explanations for the use of the metastable assumption documented. One important additional reason is that the history of the aerosols is not tracked in these models such that one cannot know whether the mutual efflorescence or deliquescence relative humidity would best characterize the conditions for crystallization. Also, consider starting a new paragraph with the metastable state discussion since this topic is slightly distinct from the thesis of the paragraph and other content in it.
   - Thank you for pointing out this missing detail, which is now included in the revised manuscript. We have also started a new paragraph with the metastable state discussion.
   - Lines 93-96: "**Another issue driving the use of the metastable state assumption in regional air quality models is the need to track the RH history of aerosols to accurately predict their phase state, due to the hysteresis of salts. Specifically, without knowing the RH history of the aerosol, it is not possible to determine whether the aerosol will exist as an aqueous solution of ions or as a crystalline salt between its efflorescence and deliquescence RH (Martin et al., 2004; Fountoukis et al., 2009).**"

3. Lines 116-8 Consider making a separate sentence with the content beginning with "which" on account of the parenthetical nature of the descriptions of parallel and vector implementations not being sufficiently clear. If that restructuring is not desired, at least make the parenthetical explanations parallel in form and each properly introduced.
   - We have restructured this section of the text.
   - Lines 132-135 of the revised manuscript have been updated: "**In this work we present HETP (HETerogeneous vectorized or Parallel), a solver based on the forward (input as gas + aerosol) metastable state algorithms of ISORROPIA II. The solver has been optimized for vector processors where similar problems for a subsystem are gathered and solved with a global convergence criterion, or parallel processors, where local case-by-case solutions to the system of equations are used to minimize processing time.**"

4. Lines 305-18 Many of the enumerated algorithmic improvements, such as the more robust solution of cubic equations, are documented with sufficient detail. One helpful addition to these enumerated algorithmic improvements would be to identify one example of their application by line number in the HETP.ftn90 file included in the Zenodo repository.
   - We have included a specific line number reference in the revised manuscript.
   - Lines 367: "**For example, the call to solve a cubic equation occurs on line 130 of subroutine 'mach_calc_hclhno3'.**"

**Grammatical Comments**

5. Line 45 Elsewhere "N" is written as an italicized variable. Consider formalizing this expression, too.
   - The variable $N$ has now been formalized in the revised manuscript.

6. Line 115 Please ensure that the "forward" nature of the solution is somehow mentioned in this helpful, governing sentence.
   - We have modified the revised manuscript as requested.

- Lines 132-133 now include a specific reference to the forward solution: "In this work we present HETP (HETerogeneous vectorized or Parallel), a solver based on the **"forward" (input precursor species as gas + aerosol)** metastable state algorithms of ISORROPIA II."

7. Line 119 It is not clear to me where the idea of the metastable state representing the "efflorescence branch" arose. To my knowledge, the mutual deliquescence relative humidities are used in ISORROPIA and insufficient information exists for the efflorescence relative humidities of these salt mixtures to treat efflorescence in a thermodynamic equilibrium model of inorganic aerosols. It would be sufficient to delete this parenthetical phrase and leave the reader to the other portions of the manuscript in which the metastable state is more accurately described.
   - We have removed "efflorescence branch" in the revised manuscript.

8. Line 159 Here and in some other cases, a ";" is missing: "initial mass adjustments, however any output" should be "initial mass adjustments; however, any output". Please revise here and in other places as needed.
   - We have revised the use of semicolons throughout the revised manuscript, including the line referenced above.

9. Line 193-4 No comma exists after the adverb at the beginning of the sentence (i.e., "Currently,") but an unnecessary comma is used before "and" in the same sentence though the conjunction is not followed by an independent clause. Elsewhere, commas are used where semicolons are needed. Please revise the use and absence of commas throughout the manuscript.
   - We have revised the use of commas throughout the revised manuscript, including the sentence referenced above.

10. Line 195-9 Four independent clauses are joined by two semicolons and one conjunction that is not preceded by a comma. Please look for run-on sentences such as these and revise as needed.
    - We have revised the manuscript to remove occurrences of run-on sentences, including the sentence referenced above.

11. Line 230 "the ZSR correlation" would be better as "the Zdanovskii-StokesRobinson (ZSR) correlation".
    - Line 262-263 in the revised manuscript: "Aerosol liquid water content in kg m$^{-3}$ air is calculated according to the **Zdanovskii−Stokes−Robinson (ZSR)** relation (Robinson and Stokes, 1965), as…"

12. Line 271 Although the code-based expression "TCl = max(TCl, 1×10-10)" will be comprehensible to those accustomed to writing Fortran, perhaps using words to express the concepts would be more appropriate for this part of the text.
    - This has been modified in the revised manuscript.
    - Lines 303-304 of the revised manuscript: "This limitation occurs since HETP does not apply the mass modification **that resets TCl to a floor value of 1×10$^{-10}$ mol m$^{-3}$**, as discussed near the start of the section."

13. Line 706 "computations into in" was likely meant to be "computations into"
    - Thank you for pointing this out. This has now been corrected in the revised manuscript.

**New references:**

Fountoukis, C., Nenes, A., Sullivan, A., Weber, R., Van Reken, T., Fischer, M., Matías, E., Moya, M., Farmer, D., and Cohen, R. C.: Thermodynamic characterization of Mexico City aerosol during milagro 2006, Atmospheric Chemistry and Physics, 9(6), 2141–2156, https://doi.org/10.5194/acp-9-2141-2009, 2009.

Martin, S. T., Hung, H.-M., Park, R. J., Jacob, D. J., Spurr, R. J., Chance, K. V., and Chin, M.: Effects of the physical state of tropospheric ammonium-sulfate-nitrate particles on global aerosol direct radiative forcing, Atmospheric Chemistry and Physics, 4(1), 183–214, https://doi.org/10.5194/acp-4-183-2004, 2004

**Other code updates**

- While updating the code to meet the Fortran standards described by CC#1 and Referee #1, we have identified some other modifications that improve the code readability and decrease the execution time of the code, as described below. These code updates reflect changes made subsequent to the submitted version of the code, and appear in the revised version, hence have been described here in more detail. Here lines starting with (-) have been removed and those starting with (+) have been added.

**(1) In CALCA2, CALCD3, CALCG5, CALCH6, CALCO7, CALCM8 and CALCP13 we have removed a section of redundant code. Specifically, the same calculation was repeated twice, with the variable 'errinlocb' being redundant since it was only used to evaluate the logical 'calain' (which can be done with 'errin'). This is only applicable to the case-by-case implementation of HETP. For example, in CALCG5 the following has been modified:**

```
(-)          errinlocb = 0
(-)          do ii = 1, 13
(-)              errinlocb = max(errinlocb, abs(gamin(ii) - gama(ii)) / gamin(ii))
(-)          end do
(-)          calain = errinlocb .ge. epsact

             errin = 0.0d0
  !  ## Test for convergence of activity coefficients
             do ii = 1, 13
                 errin = max(errin, abs((gamin(ii) - gama(ii)) / gamin(ii)))
             end do
(+)          calain = errin .ge. epsact
```

**(2) In addition to the modification above, in CALCD3 we have corrected the following lines of code in both the case-by-case and vectorized version of HETP. Note that looping over 13 activity coefficients is redundant in CALCD3 and only serves to slow down the convergence of activity coefficients.**

```
(-)          errin = 0.0d0
(-) !  ## Test for convergence of activity coefficients
(-)          do ii = 1, 13
(-)              errin = max(max(errin, abs((gamin(ii) - gama(ii)) / gamin(ii))), 0.0d0)
(-)          end do

(+)          errin = 0.0_dp
(+)          errin = max(errin, abs(gamin(4 ) - gama(4 )) / gamin(4 ))
(+)          errin = max(errin, abs(gamin(5 ) - gama(5 )) / gamin(5 ))
(+)          errin = max(errin, abs(gamin(7 ) - gama(7 )) / gamin(7 ))
(+)          errin = max(errin, abs(gamin(8 ) - gama(8 )) / gamin(8 ))
(+)          errin = max(errin, abs(gamin(9 ) - gama(9 )) / gamin(9 ))
(+)          errin = max(errin, abs(gamin(10) - gama(10)) / gamin(10))
(+)          errin = max(errin, abs(gamin(13) - gama(13)) / gamin(13))
(+)          calain = errin .ge. epsact
```

-The two modifications described above have no impact on the resultant output from the HETP solver shown in the manuscript, but they do reduce the processing time.
- As an example, consider the CALCD3 test cases obtained from GEM-MACH air-quality model, and investigated in Sect 4.2 of the manuscript. For CALCD3, the maximum absolute change in the output speciation between the originally submitted version of HETP and the updated version of HETP (including the above two modifications) is less than 0.007 %. This difference is explained by the updated methodology to solve quadratic equations (see Response #23 to Referee #1) and not by the two modifications described above. Instead, the effect of the two modifications described above is on the execution time of the code. This is reflected in the updated timing results for the case-by-case implementation shown in Table 5 of the revised manuscript (and the vectorized implementation for CALCD3).

**(3) The arrays defining the equilibrium constants in subroutine mach_hetp_main_15cases have been updated to remove unused constants – hence these arrays now have a length of 7 (instead of 15).**

**(4) Some double precision variables were assigned to numbers that were missing the "d0". This has been corrected in the code.**